# Temporal Flexibility in Spiking Neural Networks: Towards Generalization Across Time Steps and Deployment Friendliness

**Kangrui Du**[*1,3]**, Yuhang Wu**[*1]**, Shikuang Deng**[1]**, Shi Gu**[†1,2]

[1]University of Electronic Science and Technology of China
[2]Shenzhen Institute for Advanced Study, UESTC
[3]College of Computing, Georgia Institute of Technology
`kangruidu@gatech.edu, gus@uestc.edu.cn`

## Abstract

Spiking Neural Networks (SNNs), models inspired by neural mechanisms in the brain, allow for energy-efficient implementation on neuromorphic hardware. However, SNNs trained with current direct training approaches are constrained to a specific time step. This "temporal inflexibility" 1) hinders SNNs' deployment on time-step-free fully event-driven chips and 2) prevents energy-performance balance based on dynamic inference time steps. In this study, we first explore the feasibility of training SNNs that generalize across different time steps. We then introduce Mixed Time-step Training (MTT), a novel method that improves the temporal flexibility of SNNs, making SNNs adaptive to diverse temporal structures. During each iteration of MTT, random time steps are assigned to different SNN stages, with spikes transmitted between stages via communication modules. After training, the weights are deployed and evaluated on both time-stepped and fully event-driven platforms. Experimental results show that models trained by MTT gain remarkable temporal flexibility, friendliness for both event-driven and clock-driven deployment (nearly lossless on N-MNIST and 10.1% higher than standard methods on CIFAR10-DVS), enhanced network generalization, and near SOTA performance. To the best of our knowledge, this is the first work to report the results of large-scale SNN deployment on fully event-driven scenarios. Codes are available at: https://github.com/brain-intelligence-lab/temporal_flexibility_in_SNN

## 1 Introduction

As deep learning continues to evolve, the field has witnessed numerous groundbreaking advancements that have made unprecedented strides across diverse applications. However, deploying these huge neural networks on low-power edge devices presents substantial challenges. In addition to the typical solutions, including network quantization (Rastegari et al., 2016), pruning (He et al., 2017), and distillation (Hinton et al., 2015), the Spiking Neural Networks, known as one of the 3rd generation of neural networks, have emerged as a compelling candidate due to their unique bio-inspired characteristics (Fang et al., 2021; Guo et al., 2022; Yao et al., 2023). SNNs mimic the behavior of biological neurons by accumulating membrane potentials and transmitting sparse spikes, thereby circumventing the need for computationally expensive multiplications (Roy et al., 2019), which presents a promising solution for energy-efficient neuromorphic computation.

The SNN community has flourished in recent years. One of the most significant driving factors is the invention of direct training methods with time-step-based iterative neurons (Wu et al., 2018; 2019)- with an RNN-like backpropagation method, the introduction of time steps (T) successfully brings SNN training into mainstream deep learning platforms like PyTorch, which allows for fast GPU-accelerated training for large-scale SNNs.

While training at a specific time step has become a prevalent paradigm for following works, the obtained SNNs perform well only at a specific T but generalize poorly to others. This temporal inflexibility posts constraints on SNNs' deployment on neuromorphic chips, where the energy advantage is truly exploited. For example, it posts constraints on time step adjustment and thus hinders

---

[*]Equal Contribution
[†]Corresponding author

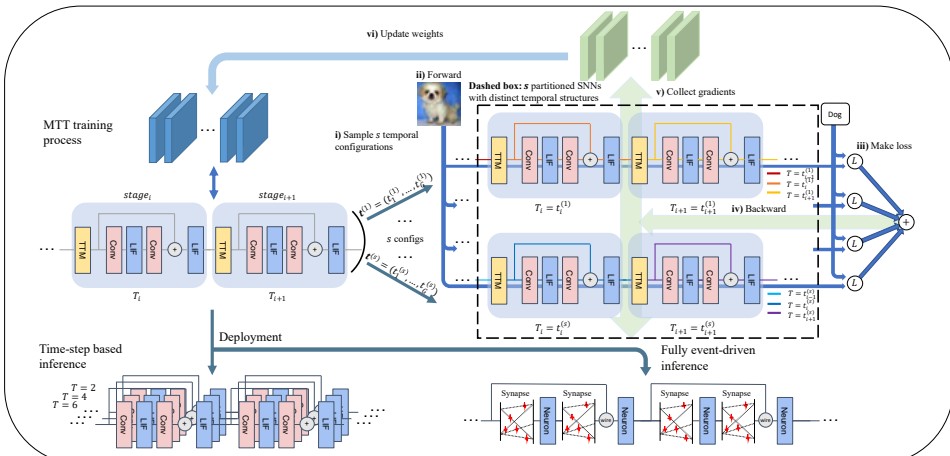

**Figure 1:** The workflow of MTT pipeline. We first partition SNN into G stages. In each iteration, we sample $s$ temporal configs $\boldsymbol{t}^{(1)}, ..., \boldsymbol{t}^{(s)} \in \mathbb{R}^G$, each assigning a set of random time steps to different stages (for $j$-th sampled config, $T_i = t_i^{(j)}$). These configurations create $s$ partitioned SNNs with distinct temporal structures, all sharing the same weights. To update the shared weights, we backpropagate the sum of the $s$ losses to obtain the gradient. Models trained with MTT exhibit temporal flexibility, which leads to their adaptation to any time step and friendliness with fully event-driven chips.

SNNs' on-device energy-performance balance by dynamic time step algorithms (Li et al., 2023d) and hardware (Li et al., 2023d). Another more conspicuous issue occurs when deploying SNNs on fully event-driven neuromorphic accelerators. Fully event-driven accelerators are ideal always-on edge platforms for SNNs with ultra-low energy consumption (Li et al., 2023b) and bio-plausibility (Richter et al., 2024). On these time-step-free platforms, all neurons operate asynchronously, and the time step here is an auxiliary for GPU-friendly training rather than a real-world hyper-parameter.

In response to these deployment issues caused by temporal inflexibility, we propose a novel training method, Mixed Time-step Training (MTT), to improve temporal flexibility of SNNs. The workflow is shown in Fig. 1. The parameters trained with this method are decoupled from temporal structures used during training and are compatible with a wide range of time steps. This not only enhances the model's compatibility with fully event-driven hardware but also enables dynamic on-chip time step adjustment and corresponding energy-performance trade-off strategies. Our main contributions are as follows:

- We identified the temporal inflexibility caused by the standard training method and its potential solution, based on which we further designed Mixed Time-step Training (MTT) to enhance the temporal flexiblility of SNNs.

- Intensive experiments are carried out on platforms including GPU-accelerated servers, neuromorphic chips, and our high-performance event-driven simulator, testifying MTT's effectiveness on static and event datasets. To our knowledge, this is the first work to evaluate large-scale SNNs in fully event-driven scenarios.

- Models trained with MTT demonstrate remarkable temporal flexibility while achieving performance comparable to other state-of-the-art approaches.

## 2 RELATED WORK

**Direct Training** The direct training approach stems from the idea that SNNs can be viewed as variant RNNs and trained using BPTT as long as the non-differentiable activation term is replaced with a surrogate gradient. Wu et al. (2018) first proposes the STBP method and trains SNNs using an ANN framework. Further, (Wu et al., 2019) and Zheng et al. (2021) suggest novel NeuronNorm and BatchNorm strategies to facilitate large-scale SNN training, respectively. Recently, the performance of SNN on neuromorphic datasets has been substantially enhanced with the advent of specially developed algorithms, including TET (Deng et al., 2022) and TCJA-SNN (Zhu et al., 2022a). On static datasets, various methods (Li et al., 2021b; Guo et al., 2022; Yao et al., 2022) are proposed to close the gap between SNNs and ANNs. Notably, Guo et al. (2022) first reported SNNs with accuracy exceeding the corresponding ANN counterpart, demonstrating the strong potential of SNNs. Moreover, the recent emergence of spike-based transformer architectures (Zhou et al., 2024; 2023; Yao

et al., 2024a;b; Zhou et al., 2022) has propelled the performance of large-scale SNNs to unprecedented levels. However, weights obtained by existing direct training methods are only applicable to a specific time step, which posts constraints for deployment and entails further fine-tuning.

**ANN-SNN Conversion** ANN-SNN conversion uses SNN firing rates to approximate the activation of ANN. Specifically, parameters are first directly copied from a pre-trained ANN to the target SNN and then fine-tuned to mimic the original ANN activation. Techniques have been proposed to reduce the minimum convertible time step, including the subtraction mechanism (Rueckauer et al., 2016; Han et al., 2020; Han and Roy, 2020), spike-norm (Sengupta et al., 2019), threshold shift (Deng and Gu, 2021), layer-wise calibration (Li et al., 2021a) and activation quantization (Bu et al., 2023). Although SNNs obtained by conversion show some temporal flexibility for large time steps (e.g., above 100), they do not exhibit temporal flexibility for ultra-low time steps. Additionally, the conversion method is unable to handle DVS datasets and can only procure SNNs with IF neurons.

**Dynamic Inference Time Step** Recent research explores inference-wise varied time steps to reduce the average inference cost by skipping steps when the network is confident enough. Li et al. (2023c) introduced SEENN, which determines the exit time step using confidence scores (SEENN-I) or a policy network (SEENN-II). Li et al. (2023a) introduced another confidence-based dynamic model, identifying the optimal confidence threshold using a Pareto front. Our models provides ideal weights for these methods due to their ability to infer at different time steps without extra fine-tuning.

**Fully Event-driven Neuromorphic Chips** Although iterative neurons successfully integrate direct training into modern backpropagation frameworks (Wu et al., 2018), clock-driven neuromorphic hardware based on this framework may not be suitable for always-on real-time devices at edge platforms due to the constant state updates even in the absence of input spikes (Dampfhoffer et al., 2022). Recently, time-step-free, fully event-driven SNN implementations have received increasing attention from researchers (Deng et al., 2024; Koopman et al., 2024) due to their ultra-low energy consumption, compatibility with real-time edge scenarios, and better similarity with biological neurons (Richter et al., 2024; Li et al., 2023b). Models trained with our method are well-suited for deployment on this fully event-driven hardware.

## 3 PRELIMINARIES

### 3.1 SPIKING NEURON MODEL

The most popular model for SNN neurons in recent studies (Wu et al., 2019; Zheng et al., 2021; Xiao et al., 2022; Deng et al., 2023; Zhou et al., 2022; Yao et al., 2024b) is the Leaky Integrate-and-Fire (LIF) model. Its dynamics can be characterized by

$$\tau_0 \frac{du}{dt} = -u + I, \, u < V_{th} \tag{1}$$

$$\text{fire a spike and } u \leftarrow R(u), \, u \geq V_{th} \tag{2}$$

where $u$ is the membrane potential, $\tau_0$ is membrane constant, $I$ is pre-synaptic input, $V_{th}$ is membrane threshold, and $R(\cdot)$ is reset function (Wu et al., 2019). For hard reset $R(u) = 0$ and for soft reset $R(u) = u - V_{th}$. The two simulation methods of these equations, the time-stepped and the event-based simulation, have given rise to two popular SNN models: the clock-driven synchronous model and the event-driven asynchronous model.

### 3.2 TIME-STEPPED SIMULATION, CLOCK-DRIVEN LIF/IF MODEL AND HARDWARE

Let's first consider Eq. 1. After applying the forward Euler method to Eq. 1, replacing $1 - dt/\tau_0$ with $\tau$, and absorbing $dt/\tau_0$ as a scaling factor into the synapse weight (Wu et al., 2019), we have

$$u(t + dt) = \tau u(t) + I(t) \tag{3}$$

where $\tau$ is a decay factor. We then incorporate fire and reset (Eq. 2) mechanism into the simulation by adding a temporary variable $v$ and have

$$v(t + dt) = \tau u(t) + I(t) \tag{4}$$

$$s(t + dt) = \Theta[v(t + dt) - V_{th}] \tag{5}$$

$$u(t + dt) = s(t + dt)R(v(t + dt)) + [1 - s(t + dt)]v(t + dt) \tag{6}$$

where $v(t)$ is the temporary variable, $s(t)$ is the output spike at time $t$, $\Theta(x)$ is the Heaviside step function where $\Theta(x) = 1$ for $x \geq 0$ and $\Theta(x) = 0$ for $x < 0$. Eq. 4-6 is the single iteration

of the time-stepped simulation. The iterative Integrate-and-Fire (IF) model is a special case of LIF where $\tau = 1$ or $\tau_0 \to +\infty$. Time-stepped simulation can be properly accelerated by GPU due to its compatibility with mainstream deep learning framework, enabling efficient training of large-scale SNNs. Clock-driven neuromorphic hardware simply mimics the iterative process of time-stepped models, which can thus be deployed on such hardware with little performance loss. Time step of clock-driven neuromorphic inference serves as a hyperparameter that controls the temporal granularity and forward time complexity, directly influencing the trade-off between performance and energy consumption. For experiments on time-stepped platforms in this paper, we use LIF neuron with $V_{th} = 1, \tau = 0.5$.

## 3.3 EVENT-BASED SIMULATION, EVENT-DRIVEN LIF/IF MODEL AND HARDWARE

Rethinking the original neuron dynamics, $I(t)$ only has discrete spikes, which can be seen as weighted Dirac delta functions. This enables a precise simulation of LIF neurons based on events. Let's start with $u < V_{th}$ again and analyze $I(t) = 0$ and $I(t) \neq 0$ separately. For cases where $I(t) = 0$, we directly solve the differential equation and have $u(t) = u(t')e^{-(t-t')/\tau_0}$ where $t'$ is the moment the last spike arrives. For $I(t) \neq 0$ cases, we let $dt \to 0$, absorb $dt/\tau_0$ into $I$ as we did in Eq. 3, and then have $du = I$ where $du$ is the instantaneous increment of $u$. The term $-dt/\tau_0 \cdot u$ is eliminated because $u$ is a finite value. Combining the two cases discussed, we have the formula

$$u(t_i) = e^{\frac{t_i - t_{i-1}}{\tau_0}} \cdot u(t_{i-1}) + I(t_i) \tag{7}$$

where $t_i$ is the timestamp of the current $i$-th spike and $t_{i-1}$ is the timestamp of the last spike (Wu et al., 2018). Similar to the time-stepped scenario, we add an intermediate variable $v$ to enable fire and reset behaviors.

$$v(t_i) = e^{\frac{t_i - t_{i-1}}{\tau_0}} \cdot u(t_{i-1}) + I(t_i) \tag{8}$$

$$s(t_i) = \Theta[v(t_i) - V_{th}] \tag{9}$$

$$u(t_i) = s(t_i)R(v(t_i)) + (1 - s(t_i))v(t_i) \tag{10}$$

where $v(t)$ is the temporary variable, $s(t)$ is the output spike at time $t$, $\Theta(x)$ is the Heaviside step function. The event-based Integrate-and-Fire (IF) model is a special case of LIF where $\tau_0 \to +\infty$. The discrete, sequential nature of events hinders parallelizing event-driven SNN training. While some event-driven methods have been explored, they either remain constrained within the time-stepped framework (Zhu et al., 2022b) or face challenges in scaling to large models due to insufficient GPU support (Engelken, 2023). Since event-driven simulation under spiking input precisely replicates the original dynamics, the industry uses time-stepped simulation to approximate event-driven behavior and directly deploys weights trained in the time-stepped framework onto fully event-driven hardware (Richter et al., 2023). In fact, Eq. 4-6 and Eq. 8-10 share the same form, and thus theoretically, **the event-driven inference can be approximated by time-stepped inference as $dt \to 0$, equivalently as time step $T \to +\infty$.** This aligns with the intuition-an sufficiently high $T$ will result in at most one event per time step, effectively mirroring the event-driven paradigm where neurons update only upon event arrival. All trainings for event-driven platforms in this paper are conducted by Speck deployment toolkit- Tonic. We use IF neurons with $V_{th} = 1$ for all event-based experiments because most of the existing fully event-driven neuromorphic chips mainly support IF for its suitability to asynchronous scenarios.

## 3.4 SURROGATE GRADIENT

The time-step-based direct training method computes gradients for parameters by spatiotemporal backpropagation (STBP) (Wu et al., 2018):

$$\frac{\partial L}{\partial \mathbf{W}} = \sum_t \frac{\partial L}{\partial \boldsymbol{s}(t)} \frac{\partial \boldsymbol{s}(t)}{\partial \boldsymbol{v}(t)} \frac{\partial \boldsymbol{v}(t)}{\partial \boldsymbol{I}(t)} \frac{\partial \boldsymbol{I}(t)}{\partial \mathbf{W}}. \tag{11}$$

When backpropagating, all terms apart from the term $\frac{\partial \boldsymbol{s}(t)}{\partial \boldsymbol{v}(t)}$ can be easily calculated. However, the term $\frac{\partial \boldsymbol{s}(t)}{\partial \boldsymbol{v}(t)} = \frac{\partial \boldsymbol{\Theta}(v)}{\partial v}$ is the derivative of the Dirac delta function and does not exist. To solve this problem, surrogate gradient(SG) is used to approximate the original gradient. For all time-stepped experiments in this work, we adopt a triangular surrogate gradient (Rathi and Roy, 2020), which can be formulated as

$$\frac{\partial \boldsymbol{s}(t)}{\partial \boldsymbol{v}(t)} = \frac{1}{h^2}\max(0, h - |V_{th} - \boldsymbol{v}(t)|), \tag{12}$$

where $h$ is a constant controlling the sharpness. In this work, we apply h=1 to most experiments. For experiments on event-driven platforms, we follow Speck handbook and use the default single exponential SG (Shrestha and Orchard, 2018) in Tonic (Lenz et al., 2021).

## 4 METHODOLOGY

### 4.1 IDENTIFYING TEMPORAL INFLEXIBILITY AND POTENTIAL SOLUTION

As discussed in Sec. 3, current mainstream methods for training SNNs, whether clock-driven or event-driven, rely on time-stepped frameworks. To avoid confusion, we refer to these conventional time-step-based training as *Standard Direct Training* (SDT). Our study identifies a critical limitation of SDT, which we named "**temporal inflexibility**": models trained with SDT perform ideally only at the same time-step setting as at training, and perform evidently lower at other time step configurations than models trained at that time step configurations specifically. To showcase this limitation, we perform experiments with ResNet18 on CIFAR100 (Krizhevsky et al., 2009). An SNN was first trained at T = 6, and its inference accuracy was evaluated across five distinct time step settings (see SDT in Tab. 1). However, the temporal structures of SNNs on platforms deployed often differ from those used during training. For example, event-driven platforms lack explicit time steps, and the time-stepped training is only an approximation to event-driven models, leading to a substantial gap with on-chip scenarios (see Sec. 3.3). In clock-driven systems, time steps may also be dynamically adjusted to meet varying energy-performance requirements (see Sec. 2 on "Dynamic Inference Time Steps"). The temporal inflexibility, however, prevents model from generalizing to temporal structure on deployment platforms and thus poses significant challenges for SNN deployment.

To alleviate temporal inflexibility, we start with a straightforward method named Naive Mixture Training (NMT). In each iteration of NMT, it randomly selects 3 time steps from a predefined range $\{1, 2, ..., 6\}$ and performs 3 forward passes. The $i$-th sampled time step $T_i$ is then used for the $i$-th forward pass of the iteration. The parameters are then updated collectively after completing all forward passes within an iteration. We test the NMT-trained model at different time steps and compare it with SDT-trained models. The results are shown at

**Table 1:** Inference accuracy of ResNet18 on CIFAR100 by naive mixture training vs. standard direct training. "SDT*": SNNs independently trained with SDT at each T. "SDT": single SNN trained at T=6 and infers at other T.

| Methods | T=2 | T=3 | T=4 | T=5 | T=6 |
|---------|-------|-------|-------|-------|-------|
| SDT | 70.08 | 72.77 | 74.17 | 75.09 | 75.63 |
| SDT* | 72.86 | 73.86 | 74.77 | 74.96 | 75.63 |
| NMT | 73.47 | 74.17 | 75.11 | 75.34 | 75.77 |

Tab. 1. Although NMT is a relatively simple method, a single NMT-trained model performs comparably to models separately trained by SDT for each time step (denoted as SDT*). This indicates that NMT effectively mitigates models' temporal inflexibility, enabling the model to generalize across different time-step configurations. We term this generalization to temporal structures beyond those used during training as **temporal flexibility**, which can be measured by its performance on unseen temporal structures compared to models trained specifically on those structures. Given the simplicity and effectiveness of NMT, we further investigate its underlying mechanisms.

### 4.2 ANALYSIS ON NAIVE MIXTURE TRAINING

**Temporal Flexibility**. As we mentioned earlier, the models trained with SDT are only optimized for a specific T. This will render overfitting towards the single temporal structure and cause poor temporal flexibility (see Tab. 1). NMT successfully mitigates this overfitting by training 6 temporal structures at the same time so that the model learns how to keep the performance with different time structures and thus gets better generalization across temporal structures. Due to the enhanced temporal flexibility, NMT-trained models can change their inference time structures with reduced performance degradation. For time-step-based SNNs, this property also detaches training time steps from the event sensor's framing time steps, which is relevant to specific application scenarios and is often too high for GPU training platforms.

**Event-driven Friendliness**. On fully event-driven hardware platforms, all neurons operate asynchronously, and the time step is no longer a hardware hyperparameter that determines the number of iterations for each inference. In such systems, the time step becomes irrelevant to the on-chip inference phase, serving only in the training phase to simulate the on-chip model's operation in a GPU-compatible manner. However, as we analyzed in Sec. 3.3 and 4.1, there is a substantial gap between time-stepped simulation and real event-driven dynamics, and temporal inflexibility caused by SDT prevents models from generalizing to event-driven temporal structures, which can be seen

as a time-stepped model whose T is infinity (see Sec. 3.3). NMT can add to model's temporal flexibility and largely decouple network outputs from the time-step-based temporal structure. This makes NMT-trained models ideal candidates for event-driven deployment.

**Network Generalization**. If SNN training falls into a local minimum point, once NMT samples a new time step that is far away from the current one, the SNN's output may change significantly. In this scenario, the new training loss may not converge, leading SNN to jump out of the local minimum point. Eventually, the SNN will be trained towards a flatter minimum point. Another perspective is that since the sampling space is 6, NMT is equivalent to training 6 similar SNNs simultaneously. This is similar to applying a new kind of dropout to the SNN, which improves the network's generalization. This is why NMT significantly improves SNN's performance when T is small. We verify our theory through experiments in Sec. 5.1 and loss landscapes in Sec. A.10.

### 4.3 MIXED TIME-STEP TRAINING

The success of NMT lies in its incorporation of diverse temporal structures during training. A straightforward idea to improve is to include more temporal structures. However, the number of temporal structures in NMT scales linearly with $T_{max}$, and excessively large $T$ cannot be trained on current GPUs. To introduce more temporal structures while keeping $T_{max}$ not increasing, we propose Mixed Timestep Training (MTT). In MTT, A normal SNN is first partitioned into stages as shown in Fig. 2. In each iteration, each stage is assigned with different time steps. By assigning stage-wise different time steps, MTT successfully expands the number of temporal structures from $T_{max}$ to $(T_{max})^G$ where G is the number of stages. We then introduce each step of MTT.

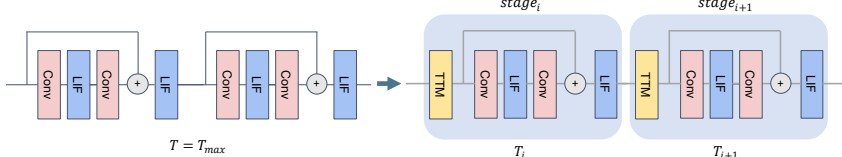

**Figure 2:** SNN partitioned for Mixed Timestep Training.

#### 4.3.1 NETWORK PARTITIONING

The first step of MTT is to divide the whole network into stages which will then be set to different time steps for each forwarding process. We denote the setting of time steps of different stages in each forwarding process of MTT by a temporal configuration vector $\boldsymbol{t} = (t_1, \ldots, t_n)$, where $t_i$ denotes the time step of the $i$-th stage and $n$ is the total number of stages. The forwarding of a partitioned SNN $S_P$ can be denoted by $S_P(\boldsymbol{x}, \boldsymbol{t})$, where $\boldsymbol{x}$ is an input and $\boldsymbol{t}$ is the temporal configuration vector.

#### 4.3.2 TEMPORAL TRANSFORMATION MODULE

We then design the inter-block communication rule between adjacent stages with different simulation time steps-temporal transformation module (TTM). When adjacent stages are of the same number of time steps, TTM becomes an identity transformation. Otherwise, TTMs can be categorized into 2 types as illustrated in Fig. 3-downsampling TTM and upsampling TTM. In downsampling

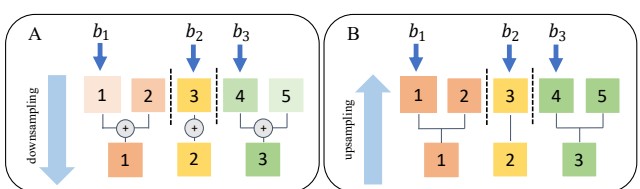

**Figure 3:** (A) Downsampling TTM when $t_{in}$=5 and $t_{out}$=3. (B) Upsampling TTM when $t_{in}$=3 and $t_{out}$=5.

TTMs, we borrow from the pooling layer and divide input time frames into $t_{out}$ groups of adjacent frames. We then sum up the frames within each group to form $t_{out}$ time frames. By contrast, the upsampling type of TTM replicates each input time frame and assigns it to all output frames in the corresponding group, using the grouping policy same as the downsampling type of TTM with $t_{out}$ frames of input and $t_{in}$ frames of output. Now, the task at hand is to identify a suitable policy to partition $l$ frames into $k$ groups, where $l \geq k$. A natural idea is to group them as evenly as possible to minimize temporal mismatch. According to this idea, we use the following policy shown in Eq. 13 where $b_i$ denotes the index of the first frame of group $i$. Explanation of our design is in Sec. A.4.

$$b_i = \lfloor \frac{(i-1) \cdot l}{k} - \varepsilon \rceil + 1, i \in \{1, ..., k\} \tag{13}$$

### 4.3.3 Mixed Time-step Training

**Overall MTT Framework** With TTM and a partitioned network, we can now develop mixed time-step training (MTT). Mathematically, the goal of MTT is to minimize loss for all temporal structures:

$$\mathcal{L}_{MTT(overall)} = \sum_{k=1}^{N} \sum_{\boldsymbol{t} \in \{T_{min}, ..., T_{max}\}^G} \mathcal{L}(S_P(\boldsymbol{x}_k, \boldsymbol{t}), \boldsymbol{y}_k) \tag{14}$$

where $\mathcal{L}$ is any loss function, $N$ is dataset size, $G$ is the number of stages, $T_{min}$ and $T_{max}$ are the minimum and maximum time steps respectively, $S_P$ is a partitioned SNN, and $\boldsymbol{t}$ is any possible temporal configuration vector. Since directly optimizing the overall loss is too expensive, we sample $s$ vectors $\boldsymbol{t}^{(1)}, \ldots, \boldsymbol{t}^{(s)} \in \{T_{min}, \ldots, T_{max}\}^l$ for each iteration and optimize the estimated loss function instead:

$$\mathcal{L}_{MTT} = \sum_{k=1}^{B} \sum_{\boldsymbol{t} \in \{\boldsymbol{t}^{(1)}, ..., \boldsymbol{t}^{(s)}\}} \mathcal{L}(S_P(\boldsymbol{x}_k, \boldsymbol{t}), \boldsymbol{y}_k) \tag{15}$$

where $B$ is batch size. To better illustrate our method, the training pipeline of one epoch is detailed in Algo.1. In this work, $T_{min}$ is set to 1 for all experiments.

---

**Algorithm 1** Mixed time step training for one epoch

---

**Input:** SNN model $S_P$; training dataset; training iteration $I$; sample number $S$ in one iteration;
stage number $G$; minimum and maximum time step $T_{min}, T_{max}$

1: **for** all $i = 1, 2, \ldots, I$-th iteration **do**
2:     Get the training data $\boldsymbol{x}_i$ and labels $\boldsymbol{y}_i$
3:     **for** all $s = 1, 2, \ldots, S$-th sample **do**
4:         Sample a vector $\boldsymbol{t}^{(s)}$ with $G$ numbers in the range $[T_{min}, T_{max}]$
5:         Calculate the loss function $\mathcal{L}(S_P(\boldsymbol{x}_i, \boldsymbol{t}^{(s)}), \boldsymbol{y}_i)$
6:         Backpropagation and collect the gradient
7:     **end for**
8:     Update the model weights with collected gradients
9: **end for**

---

**BN Layer Calibration** MTT implemented with the standard BN technique suffers significant accuracy degradation. When training with mixed time steps, drastic structural changes will cause intensive variations in batch statistics. Therefore, the running mean and variance calculated during training are inaccurate for models trained with MTT. To address this problem, we either lock BN layers (when fine-tuning, Sec. A.3) or calibrate BN statistics similar to Yu and Huang (2019) from a few training batches after optimizing and fixing weights (when training). In our experiments in Sec. 5.2, correcting BN statistics with as few as 10 batches proved sufficient. Therefore, we used this setting in all experiments. We also found that the BN statistics of $T_{max}$ apply to other time steps. This saves us extra calibrations when switching to a different inference T (details in Sec. A.6).

### 4.4 Tests on Fully Event-driven Scenarios

To verify the event-driven friendliness of our TFSNN, we employed Synsense Speck2e as the testing platform. The Speck series chips are among the most advanced real-time, fully event-driven neuromorphic chips available today, boasting extremely low power consumption and latency (Richter et al., 2023; Li et al., 2023b). However, due to their limited size, Speck cannot support the mainstream backbones used for DVS datasets in recent studies (e.g. VGGSNN). Therefore, we developed an easy-to-use parallelized event-driven chip software simulator and aligned it with Speck on small datasets. To quantify the mismatch between a given output and the real hardware output, we define spike difference (SD) as

$$SD(\boldsymbol{s}_0, \boldsymbol{s}) = \frac{\sum_{i=0}^{N_f - 1} |s_0[i] - s[i]|}{\sum_{i=0}^{N_f - 1} s_0[i]} \tag{16}$$

where $\boldsymbol{s}_0$ is the real hardware output spikes, $\boldsymbol{s}$ is the output spikes to compare with, $s_0[i]$ denotes the total spike counts of the $i$-th neuron, and $N_f$ is the dimention of output. We then used this simulator to test TFSNN on larger datasets and mainstream backbones (see Sec. 5.1). To the best of our knowledge, our work is the first to report the performance of large-scale datasets and models on a fully event-driven scenario.

## 5 EXPERIMENTS

In this section, we first conduct validation experiments for analyses in Sec. 4.2. These experiments aim to examine if the models trained with MTT improves temporal flexibility, event-driven friendliness, and network generalization as we claim. Then, a comparison between our method and other current training methods is made to demonstrate the effectiveness of our method in terms of temporal flexibility and model performance. Finally, well-designed ablation studies are carried out to prove the effectiveness of network partitioning and BN calibration, the two main components of our method. The datasets involved in this work include static datasets like CIFAR10, CIFAR100 (Krizhevsky et al., 2009), and ImageNet (Deng et al., 2009), and event-based datasets such as CIFAR10-DVS (Li et al., 2017) and N-Caltech101 (Orchard et al., 2015). We also tested our method on sequence task and audio task (see Sec. A.12). The model structures used in this paper are ResNet-18 (He et al., 2016), ResNet-19 (Zheng et al., 2021), ResNet-34 (He et al., 2016), VGG.

### 5.1 VALIDATION EXPERIMENTS

**Temporal Flexibility Across Time Steps** We first test our models under a wider range of inference time steps on the datasets and network structures. On static datasets, inputs are repeated for T times as previous works where T is the inference time step. MTT-trained models perform fairly well at different time steps as shown in Tab. 2. On DVS datasets, input events are split into T frames according to their timestamps for time-stepped training and inference. See training details in Sec. A.3. We find the model's temporal flexibility is significantly enhanced by MTT especially at high T as shown in Fig. 4 and inference at high T is similar to event-driven scenario (Sec. 3.3), while SDT and TET both fail to generalize across time steps. We then further compare our method with recent ANN-SNN conversion methods in Tab. 3 to demonstrates the temporal flexibility brought by MTT. Although these conversion methods involve post-conversion finetuning at inference T and MTT doesn't, MTT still outperforms them significantly at T=1 and T=2 while remaining comparable to them for higher time steps. For a fair comparison, we adopt the same data augmentation policy as these methods. To further demonstrate the temporal flexibility that MTT brings, and to show how temporal flexibility benefits clock-driven deployment as we claimed, we combine SEENN (Li et al., 2023c) with ResNet19 trained by MTT on CIFAR10 (Tab. 4). Since the time step in SEENN-I is dynamically decided by confidence scores, the T ultimately contains decimals, which stand for the average number of time steps needed for each sample in the test set. We observed a performance boost under the same average inference T when compared with their reported results trained by TET, testifying the suitability of MTT-trained models to dynamic time step inference methods.

**Table 2:** Accuracy of different inference time steps.

| Dataset | Method | Backbone | T=2 | T=3 | T=4 | T=5 | T=6 |
|---|---|---|---|---|---|---|---|
| CIFAR100 | **MTT** | ResNet-19 | 80.35 | 81.14 | 81.51 | 81.73 | 81.98 |
| CIFAR10 | **MTT** | ResNet-19 | 96.20 | 96.62 | 96.75 | 96.76 | 96.84 |
| ImageNet | **MTT** | ResNet-34 | 65.23 | 67.58 | 67.54 | 68.02 | 68.34 |

**Table 3:** Compare with SOTA ANN-SNN conversion methods on CIFAR100, ResNet18. $T_{max} = 6$ is used for MTT.

| Method | T=1 | T=2 | T=4 | T=8 | T=16 | T=32 | T=64 |
|---|---|---|---|---|---|---|---|
| QCFS Bu et al. (2023) | - | 70.29 | 75.67 | 78.48 | **79.48** | **79.62** | **79.54** |
| SlipReLU (Jiang et al., 2023) | 71.51 | 73.91 | 74.89 | 75.40 | 75.41 | 75.30 | 74.98 |
| MTT | **72.09** | **76.54** | **78.47** | **78.90** | 79.17 | 79.25 | 79.42 |

**Table 4:** Combine MTT with SEENN.

| Method | T=1.20 | T=1.09 |
|---|---|---|
| SEENN-I (Li et al., 2023c) | 96.38 | 96.07 |
| SEENN-I + MTT | **96.58** | **96.08** |

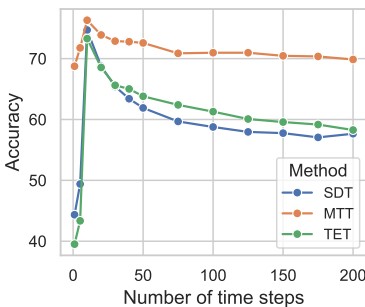

**Figure 4:** MTT-trained model has a universally high performance across different number of time steps for VGGSNN on CIFAR10-DVS. $T = 10$ is used for SDT, TET and $T_{max} = 10$ is used for MTT.

**Event-driven Friendliness** Since MTT-trained models have improved temporal flexibility, they are more suitable for deployment on fully event-driven chips as we claimed. To prove this, we deploy and test our MTT-trained models on event-driven neuromorphic systems. We first train two networks on NMNIST using SDT and MTT respectively, and then deploy them directly on Speck2e Devkit (Richter et al., 2023). We then develop an easy-to-use software simulator for event-driven chips to test MTT on large-scale datasets and models. Experiments show that our simulator accurately mimics the chip behavior. On the NMNIST dataset, the spike difference (SD, see Eq. 16) between the simulator's output and the actual on-chip output is only **3.92%**, comparable to the

2.81% SD between two identical model tests on the same chip, much lower than the 28.77% SD between time-step-based inference and the on-chip output. Our supportive results in Tab. 5 on various datasets and backbones strongly demonstrate the event-driven friendliness of MTT-trained models. In this section, our models show slightly lower PyTorch test accuracy compared to models with similar backbones because they are bias-free to be deployed on chips. Fully event-driven chips like Speck eliminate time-step-wise operations, including clocked bias addition, making them extremely energy-efficient and ideal for always-on scenarios. See Sec. A.3 for more training details.

**Table 5:** Comparison of model performance across different datasets and methods.

| Dataset | Backbone | Param Size | Method | Torch | Simulator | Speck |
|---|---|---|---|---|---|---|
| N-MNIST | 3C1FC(W16) | 6160 | MTT (Tmax=10) | 99.16 | **98.56** | 98.57 |
| | | | SDT (T=10) | 98.09 | 93.07 | 92.77 |
| DVS-Gesture | 4C2FC(W32) | 48768 | MTT (Tmax=40) | 88.28 | **81.82** | - |
| | | | SDT (T=40) | 88.67 | 80.68 | - |
| CIFAR10-DVS | VGGSNN | 9228416 | MTT (Tmax=10) | 75.2 | **58.5** | - |
| | | | SDT (T=10) | 74.7 | 48.4 | - |

**Network Generalization** As mentioned earlier in the Sec. 4.2, our method makes the network parameters robust against network structure changes and adds to the models' generalization. We further verified this through experiments with ResNet-18 on CIFAR100. A common method to measure the model's generalization is noise injection. First, we randomly inject Gaussian noise $\mathcal{N}(0, \sigma^2)$ to weights, where $\sigma^2$ is the variance of the noise. For each $\sigma^2$, we run

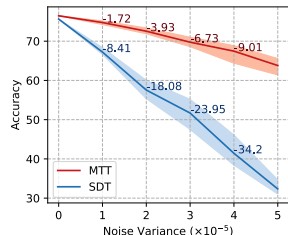 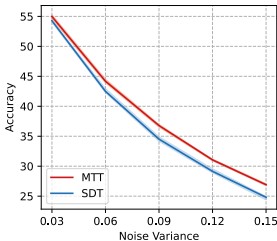

**Figure 5:** Accuracy of models with weights injected with Gaussian noise

**Figure 6:** Accuracy of models with inputs injected with Gaussian noise

the experiment 5 times and plot the mean, maximal, and minimal accuracy in Fig. 5. Results show that the weights trained by MTT are more robust against noise. Then, we inject Gaussian noise into the inputs instead, and also run the experiment 5 times each $\sigma^2$, the results are shown in Fig. 6. In addition, to solidify the conclusion, we also inspect the generalization in other metrics (see Sec. A.11). All the experiments indicate that the model obtained by MTT performs better generalization.

## 5.2 ABLATION STUDY

**Evolution from SDT to NMT to MTT** Our improvement to NMT mainly lies in dividing networks into time-step-different stages. In this section, we test MTT with different partition granularity on CIFAR100 with ResNet-18 to validate the effectiveness of network partitioning. We define granularity constant g as the number of blocks per stage. NMT can be now seen as a special case where g=8 (there are 8 blocks in ResNet-18). Then, we train 4 SNNs with MTT and g=1,2,4,8 respectively, and another single SNN with SDT for comparison. Finally, we assess test accuracy for each model with T=2,3,4,5,6. The results are shown in Fig. 7 (A). As expected, with g continued to reduce and more temporal structures added to the optimization space, the overall performance is generally improved.

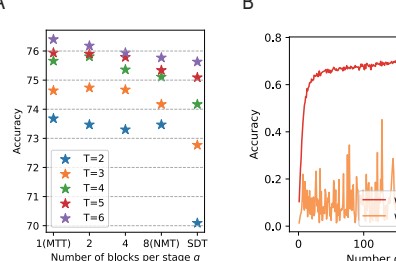

**Figure 7:** (A) Accuracy of ResNet-18 with different granularity where $g$=8 denotes NMT and $g$=1 denotes MTT. SDT denotes a single model trained at T=6 and tested across T=2,3,4,5. (B) Training ResNet-18 on CIFAR100 and tracing the test accuracy with and without BN calibration.

**Effectiveness of BN Calibration** In this section, we studied the necessity of BN calibration. We first trained two ResNet-18 on CIFAR100 and tracked their accuracy, one with BN calibration on 10 batches before testing and one simply using running means and variances. The results are shown in Fig. 7 (B). Our experiment indicates that without correct BN statistics, the model suffers huge accuracy degradation and that BN calibration effectively ameliorates the degradation.

**Table 6:** Compare with existing works on static image datasets. † denotes introducing additional floating-point multiplications

| Dataset | Model | Methods | Architecture | TimeStep | Accuracy |
|---|---|---|---|---|---|
| CIFAR10 | Guo et al.(Guo et al., 2022) | InfLoR-SNN | ResNet-19 | 6 | 96.49±0.08 |
| | | | | 4 | 96.27±0.07 |
| | | | | 2 | 94.44±0.08 |
| | Deng et al.(Deng et al., 2022) | TET | ResNet-19 | 6 | 94.50±0.07 |
| | | | | 4 | 94.44±0.08 |
| | | | | 2 | 94.16±0.03 |
| | Yao et al.(Yao et al., 2022) | GLIF† | ResNet-19 | 6 | 95.03±0.08 |
| | | | | 4 | 94.85±0.07 |
| | | | | 2 | 94.44±0.10 |
| | **Our Method** | MTT | ResNet-19 | 6 | **96.84**±0.03 |
| | | | | 4 | **96.75**±0.04 |
| | | | | 2 | **96.20**±0.07 |
| CIFAR100 | Li et al.(Li et al., 2021b) | Dspike | ResNet-18 | 6 | 74.24±0.10 |
| | | | | 4 | 73.35±0.14 |
| | Guo et al.(Guo et al., 2022) | InfLoR-SNN | ResNet-19 | 6 | 79.51±0.11 |
| | | | | 4 | 78.42±0.09 |
| | Deng et al.(Deng et al., 2022) | TET | ResNet-19 | 6 | 74.72±0.28 |
| | | | | 4 | 74.47±0.15 |
| | Yao et al.(Yao et al., 2022) | GLIF† | ResNet-19 | 6 | 77.35±0.07 |
| | | | | 4 | 77.05±0.14 |
| | **Our Method** | MTT | ResNet-19 | 6 | **81.98**±0.03 |
| | | | | 4 | **81.51**±0.04 |
| ImageNet | Zheng et al. (Zheng et al., 2021) | STBP-tdBN | ResNet-34 | 6 | 63.72 |
| | Deng et al. (Deng et al., 2022) | TET | ResNet-34 | 4 | 64.79 |
| | Fang et al. (Fang et al., 2021) | SEW† | SEW-ResNet-34 | 4 | 67.04 |
| | Chen et al. (Chen et al., 2023) | MPSNN† | DSNN-34 | 4 | 67.52 |
| | | FSNN | FSNN-34 | 4 | 66.45 |
| | **Our Method** | MTT | ResNet-34 | 6 | **68.34** |
| | | | | 4 | **67.54** |

**Table 7:** Compare with existing works on DVS datasets. †denotes introducing additional floating-point multiplications

| Dataset | Model | Methods | Architecture | T | Accuracy |
|---|---|---|---|---|---|
| CIFAR10-DVS | Yao et al. (Yao et al., 2022) | GLIF† | 7B-wideNet | 16 | 78.10 |
| | Guo et al. (Guo et al., 2022) | InfLoR-SNN | ResNet-19 | 10 | 75.50±0.12 |
| | Zhu et al. (Zhu et al., 2022a) | TCJA-SNN† | VGGSNN | 10 | 80.7 |
| | Deng et al. (Deng et al., 2022) | TET | VGGSNN | 10 | 83.17±0.15 |
| | **Our Method** | MTT | ResNet-18 | 10 | 82.8±0.54(**83.5**) |
| N-Caltech101 | Kim et al. (Kim and Panda, 2021) | SALT | VGG11 | 20 | 55.0 |
| | Li et al. (Li et al., 2022) | NDA | VGG11 | 10 | 78.2 |
| | Zhu et al. (Zhu et al., 2022a) | TCJA-SNN† | VGGSNN | 14 | 78.5 |
| | **Our Method** | MTT | ResNet-18 | 10 | **81.74**±0.73(**82.32**) |

## 5.3 Comparison to Existing Works

While our work mainly focuses on improving the temporal flexibility of networks, the models trained by MTT maintain a performance on par with other SOTA methods. See Sec. A.2 for training details. Here, we compare the accuracy of models trained by MTT with existing works. Remarkably, to demonstrate the superior temporal flexibility of MTT-trained networks, for one backbone and one dataset in the table, we trained only once and tested for all different T. For all experiments, we apply the sampling number $s = 3$ unless otherwise specified. The results of static and neuromorphic datasets are provided in Tab. 6 and Tab. 7. We repeat the experiment three times to report the mean and standard deviation (see Sec. A.2 for details).

## 6 Conclusion

In this paper, we have identified "temporal inflexibility", a side effect caused by the prevailing training paradigms. This issue, which has not yet received sufficient attention, can lead to significant challenges when deploying GPU-trained models on neuromorphic devices. To respond to the deployment issues, we propose a novel training method, Mixed Time-step Training, to enhance SNN's temporal flexibility. We conduct intensive experiments on GPU, neuromorphic chips, and event-driven simulator to prove its effectiveness. The results indicate that MTT significantly enhances temporal flexibility, improves compatibility with event-driven systems while maintaining performance comparable to state-of-the-art approaches. The idea of introducing diverse training temporal structures in MTT can also be applied to other training forms, such as randomly accumulating input frames or adjusting time steps in online training (Xiao et al., 2022; Bellec et al., 2020). We believe our methods pave a path to the promising future of nearly lossless event-driven hardware deployment and will inspire other impressive designs. Finally, we sincerely hope that our work will draw more academic attention to the deployment issues of SNNs on event-driven neuromorphic chips.

## 7 ACKNOWLEDGEMENTS

This project is supported by NSFC Key Program 62236009, Shenzhen Fundamental Research Program (General Program) JCYJ 20210324140807019, NSFC General Program 61876032, and Key Laboratory of Data Intelligence and Cognitive Computing, Longhua District, Shenzhen.

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

# A  APPENDIX

## A.1  EXPERIMENTS ON VGG STRUCTURES

We evaluated our method on the CIFAR100 dataset using VGG architectures, besides ResNets. We treated each layer as a stage in the VGG series. During experimentation, we observed that VGG16 with three fully connected (fc) layers could not be trained effectively using the standard direct training approach (its accuracy remained limited at 1%). To tackle this issue, we merged the last three fc layers of VGG16 into one and named the resulting architecture VGG14. We set $T_{max} = 5$, $s = 3$, and computed the mean and standard deviation of three runs. We used the SGD optimizer to train the model, with a learning rate of 0.1, a weight decay of 0.0005, and a batch size of 256. The results, presented in Tab. 8, indicate the effectiveness of our approach on VGG structures.

**Table 8:** Accuracy of VGG on CIFAR100

| Methods | Model | T=2 | T=3 | T=4 | T=5 |
|---------|-------|-----|-----|-----|-----|
| MTT | VGG14 | 73.53±0.09 | 74.52±0.07 | 75.27±0.14 | 75.72±0.10 |
| InfLoR-SNN | VGG16 | - | - | - | 71.56±0.10 |

## A.2  TRAINING DETAILS FOR NON-EVENT-DRIVEN EXPERIMENTS

**CIFAR** The CIFAR10/CIFAR100 dataset comprises 50K training images and 10K test images with a 32×32 pixel resolution. For CIFAR100, we train a ResNet-19 using the MTT pipeline for 300 epochs with a batch size of 256 and a $T_{max}$ of 6. Following the practice in GLIF (Yao et al., 2022), the last 2 fully connected layers of ResNet-19 are replaced with a single fully connected layer. We employ the SGD optimizer with a weight decay of 0.0005 and a learning rate of 0.1 cosine decayed to 0. To make a fair comparison with the state-of-the-art (SOTA) work (Li et al., 2021b; Guo et al., 2022; Yao et al., 2022), AutoAugment (Cubuk et al., 2018) and Cutout (DeVries and Taylor, 2017) are applied to both CIFAR10 and CIFAR100 datasets. However, these augmentation techniques are only used for comparative experiments and temporal flexibility experiments, and not for other experiments.

**ImageNet** ImageNet (Deng et al., 2009) contains more than 1280k training images and 50k test images. We use the standard data processing flow to crop each image to a size of 224×224. We deploy the ResNet-34 structure, however, with the removal of the first max-pooling layer and changing the stride of the first basic block from 1 to 2 (Zheng et al., 2021; Yao et al., 2022). We train the model for 160 epochs with a batch size of 512 and a $T_{max}$ of 6. We utilize the AdamW optimizer with a weight decay of 0.02 and a learning rate of 0.004 cosine decayed to 0.

**DVS-Dataset** There are two training settings on DVS-datasets in this work, one is for comparison with existing state-of-the-arts, and another is for all other experiments on event-driven datasets where we ensure the models are deployable on fully event-driven chips. However, the data processing of these two settings are the same. The event datasets involved in this paper, NMNIST, DVS-Gesture, CIFAR10-DVS and N-Caltech101, are neuromorphic datasets widely used in SNN experimentation. For NMNIST, we don't do any data augmentation and remain the resolution. For DVS-Gesture, we perform random horizontal flip and roll the frames up to 20 pixels. For CIFAR10-DVS and N-Caltech101, we divide the dataset into a 9:1 ratio, which, similar to previous work (Li et al., 2021b; Deng et al., 2022), we resize to 48×48. For both these datasets, we adopt a random horizontal flip and rotate the frames up to 5 pixels as augmentation techniques. We employ the additional temporal inversion policy (Shen et al., 2023) uniquely for N-Caltech101.

We then introduce the settings of SOTA comparison experiments (see next section for event-driven settings), which is for time-stepped inference and only used in Sec. 5.3. Following previous works, we merge all events to form ten frames. The optimizer we choose is SGD, with a weight decay of 0.0005, and a learning rate of 0.1, which we cosine decay to 0. For both datasets, we use a $T_{max}$ of 10, a batch size of 50, and train standard ResNet-18 model for 300 epochs. We take only the first $t$ frames of the ten frames where $t$ denotes the time step of the input stage, to feed into the network.

## A.3  TRAINING DETAILS FOR EVENT-DRIVEN EXPERIMENTS

Here, we provide the settings used by all other experiments on event-driven datasets than SOTA comparison. As we mentioned, the settings introduced in the last section is only applicable to SOTA

comparions (experiments in Sec. 5.3). All other experiments related to event datasets (e.g. Sec. 5.1) use the settings described in this section, including experiments in Fig. 4 and Tab. 5

We carefully learned Synsense's documentation. To obtain Spiking Neural Networks (SNNs) more suitable for deployment on asynchronous chips, we utilized a soft reset mechanism and multistep-IF neurons during training (neurons emit mem/Vth spikes per event to simulate multiple event transmissions and generations). Note that, however, the neurons realistically employed on our simulator and Speck chip are still IF neurons.

Since asynchronous event-driven chips do not support linear or convolutional layers containing biases, and networks with Batch Normalization (BN) layers inevitably introduce bias terms after absorbing BN into the convolutional layers, we adopted a more indirect method to obtain deployable networks. For 3C1FC(W16) (xCyFC(Wz) denotes VGG-like structure with x convolution layers, y fully connected layers, and maximum convolution channel z) on N-MNIST, For larger and deeper networks like 4C2FC(W32) for DVS-Gesture and VGGSNN for CIFAR10-DVS, we first train with networks that include BN layers with SDT to establish a baseline model, then absorb BN into the convolutional layers, remove the bias, and further fine-tune to achieve a network without bias terms. MTT is applied to the bias-removal finetuning process with BN locked.

For NMNIST, our final layer is an IF voting layer; for DVS-Gesture and CIFAR10-DVS, we find it crucial to replace the final layer with a membrane potential voting layer.

## A.4 DESIGN TTM GROUPING POLICY

We previously stated our policy's objective is to partition $l$ frames into $k$ groups ($l \geq k$) as evenly as possible. In this section, we mathematically interpret the design. We will start by describing the grouping process in a different manner. The $l$ frames are viewed as $l$ adjacent intervals of length 1 over the rational number domain with the $i$-th frame starting at $i-1$ and ending at $i$. We define $c_i$ as the boundary between group $i$ and group $i-1$. Here, $i$ ranges from 1 to $k$, and $c_1$ is 0. Ideally, $c_i = (i-1) \cdot l/k$ is set to group frames most evenly. Nevertheless, this strategy produces non-integer $c_i$, which results in atomic frames' division when $l$ is not a multiple of $k$. To solve this problem, we retreat and set $c_i$ to the nearest integer and get

$$c_i = \lfloor \frac{(i-1) \cdot l}{k} - \varepsilon \rceil, \tag{17}$$

where $\varepsilon$ is a small constant used to determine $c_i$ when the distances to the closest two integers are equal. As $b_i$ must be the frame directly following the boundary $c_i$ ($b_i = c_i + 1$), we obtain Eq. 13.

## A.5 DERIVATION OF THE BACKPROPAGATION FORMULA FOR LIF

In this section, we derive the backpropagation formula for LIF from the forwarding formula. We derive $\partial u(t)/\partial v(t)$ from Eq. 4 first:

$$\frac{\partial u(t)}{\partial v(t)} = 1 - s(t) - v(t) \cdot \frac{\partial s(t)}{\partial v(t)}. \tag{18}$$

Then, we consider the derivation of $\partial u(t)/\partial v(t-1)$.

$$\frac{\partial u(t)}{\partial v(t-1)} = \frac{\partial u(t)}{\partial v(t)} \frac{\partial v(t)}{\partial u(t-1)} \frac{\partial u(t-1)}{\partial v(t-1)} = \tau \frac{\partial u(t)}{\partial v(t)} \frac{\partial u(t-1)}{\partial v(t-1)}. \tag{19}$$

By combining multiple Eq. 19, we get

$$\frac{\partial u(t)}{\partial v(t-n)} = \tau^n \prod_{i=t-n}^{t} \frac{\partial u(i)}{\partial v(i)}. \tag{20}$$

Finally, we get the complete expression for $\partial s(t)/\partial I(t-n)$ as follows

$$\begin{aligned} \frac{\partial s(t)}{\partial I(t-n)} &= \frac{\partial s(t)}{\partial v(t)} \frac{\partial v(t)}{\partial u(t-1)} \frac{\partial u(t-1)}{\partial v(t-n)} \frac{\partial v(t-n)}{\partial I(t-n)} \\ &= \frac{\partial s(t)}{\partial v(t)} \cdot \tau^n \prod_{i=t-n}^{t-1} [(1-s(i)) - v(i) \cdot \frac{\partial s(i)}{\partial v(i)}] \end{aligned} \tag{21}$$

### A.6 T = T$_{\text{MAX}}$ BN STATISTICS VS. RECALCULATED BN STATISTICS

As previously mentioned, we discovered that the BN statistics of the T=T$_{\text{max}}$ network can be applied to other temporal structures with uniform T across blocks. In this section, we provide experimental verification for this observation. Our experiment involves testing the accuracy of one of our trained ResNet-19 models with two distinct BN layer information approaches. The first approach utilizes the statistics of the T=T$_{\text{max}}$, while the second approach recalculates the BN statistics individually for each time step T. For the latter approach, we calibrate the BN layer three times and report the average accuracy. Our experimental results demonstrated in Tab. 10 show that the mean and variance calculated at T=T$_{\text{max}}$ is applicable directly to other T values. Therefore, we can utilize the BN statistics of T=T$_{\text{max}}$ for other T directly, which saves the time required to calibrate the BN layers for other T values.

**Table 9:** Accuracy given sampling number $s$ and training epochs $e$.

| Sampling Num | $e \times s$ | | | |
|---|---|---|---|---|
| | 300 | 600 | 900 | 1200 |
| $s = 1$ | **75.61** | 76.13 | 76.23 | 75.78 |
| $s = 2$ | 75.53 | 76.37 | 76.31 | 76.36 |
| $s = 3$ | 75.25 | **76.45** | **76.47** | **76.44** |
| $s = 4$ | 74.81 | 75.74 | 76.41 | 76.42 |

**Table 10:** Test accuracy of a single model with two kinds of BN statistics.

| Method | TimeStep | | | |
|---|---|---|---|---|
| | 2 | 3 | 4 | 5 |
| T=T$_{\text{max}}$ stat | 80.21 | 81.06 | **81.51** | 81.82 |
| Recalculated stat | 80.21 | **81.23** | 81.44 | **81.85** |

### A.7 IMPACT OF DIFFERENT SAMPLING NUMBER AND TRAINING EPOCHS

In most previous experiments, we employed sampling number $s = 3$. In this section, we experiment with varying values of $s$, assess their effects at different epochs, and explain why we chose $s = 3$. We train ResNet-18s with $T_{max} = 6$ on CIFAR100 with varied $s$ and list their test accuracy at T=6. The results are displayed in Tab. 9. When $e \times s$ is constrained, the model requires more epochs to converge, necessitating a lower $s$. However, if trained across a sufficient number of epochs, sampling of $s$ structurally diverse networks can smooth the optimization of network parameters and improve performance. Specifically, we find $s = 3$ performs well and adopt $s = 3$ for most of the experiments.

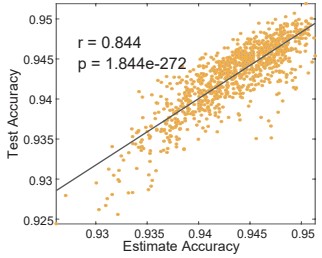

**Figure 8:** Correlation curve for estimate accuracy and test accuracy on CIFAR10

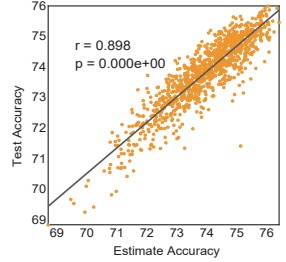

**Figure 9:** Correlation curve for estimate accuracy and test accuracy on CIFAR100

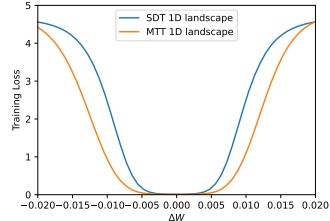

**Figure 10:** The 1D landscapes of ResNet18 trained by SDT/MTT on CIFAR100.

### A.8 PARTITIONED SNN ACCURACY ESTIMATION FOR DIFFERENT TEMPORAL CONFIGURATIONS ON CIFAR10/100

As MTT divides SNN into stages of different T when training, we are interested in the performance of each possible temporal configuration $t$. Although the stage-wise different time steps are only applied to the MTT training process to improve the temporal flexibility of SNNs and are not for inference, the relationship between $t$ and inference performance can inspire SNN structure designs. However, it is impossible to test all possible situations directly (e.g., there are $6^8$ cases for ResNet-18). Therefore, we propose the following hypotheses to estimate the accuracy of each $t$: 1) The expressive power of each stage contributes differently and positively to the final network accuracy. 2) The expressive power of each stage is related to the information content of its selected time T, such as $K\sqrt{\log_2 T}$, where the square root is due to the information content of a spike sequence with

time step T cannot exceed $\log_2 T$ because the arrangement of spikes is regular, e.g., tending to be uniformly distributed. Then we assume that the accuracy equation of different temporal configurations is $\sum_{i=1}^{I} K_i \sqrt{\log_2 T_i} + c$, where the $K_i$ is the contribution of each block, and c is a constant. Our experiment demonstrates that we only need to sample a very small number of $\boldsymbol{t}$s to infer the parameters of the equation, and then able to estimate accuracies of all temporal configurations with this equation.

Here, we randomly sample 18 temporal configuration vectors (different time step combinations) and obtain their test accuracy for solving the hypothesis equation in Sec. A.8. The weight parameters we obtained are $\{0.93, 0.53, 0.59, 0.67, 1.22, 0.48, 1.36, 0.18\}$, and the constant value $c$ is 67.11. This result supports our hypothesis 1), which suggests that all blocks' time step increment positively contributes to the accuracy of the final network. Some blocks, such as blocks 1, 5, and 7, have a greater contribution. Then, we resample 1000 configurations and validate their estimated accuracy and their test accuracy. The result (Fig. 7 (A)) shows that our method can effectively predict the actual testing accuracy of different temporal configurations. Finally, we use the spike frequency and accuracy estimation to build a combinatorial optimization equation for searching the optimal configuration (see Sec A.9 for detail). For example, by setting the energy cost that is lower than default (the time step of all blocks is 3), we discover the optimal combination of block time steps is $\{3, 2, 2, 3, 5, 3, 6, 2\}$. The selected configuration acquires an accuracy of 75.38%, which is 0.62% higher than the default.

In addition to CIFAR100, we also conducted experiments with ResNet-18 on CIFAR10/100. We randomly sample 18 temporal configurations as usual and solve the equation in Sec. A.9. The weight parameters are $\{0.0049, 0.0028, 0.0017, 0.0037, 0.0037, 0.0004, 0.0005, 0.0017\}$, and the constant value $c$ is 92.13. Notes that block 1,4,5 have a higher contribution, and the 1,5 blocks are also highly weighted on CIFAR100, which may imply that the weights are partly related to the network structure. We then resample 1000 temporal configurations and plot their estimate accuracy and test accuracy on CIFAR10 in Fig. 8 and CIFAR100 in Fig.9, respectively. We also use the aforementioned combinatorial optimization strategy to search the optimal temporal configuration under the energy consumption of T=3 and find $\{4, 2, 2, 3, 5, 2, 2, 4\}$, which achieves an accuracy of 94.70%, 0.21% higher than its T=3 counterpart.

## A.9 DETAILS OF COMBINATORIAL OPTIMIZATION

As previously mentioned, the accuracy formula of different temporal structures is given by the expression $\sum_{i=1}^{I} K_i \sqrt{\log_2 t_i} + c$, where $K_i$ represents the contribution of each block, $I$ is the number of blocks, and $c$ is a bias. We randomly select 18 distinct temporal configurations ($\boldsymbol{t}$) and evaluate their accuracies on the test set, resulting in 18 pairs of temporal configurations and their corresponding accuracies. Using the least squares method, we compute the values of $K_i$ and $c$ from the collected data. Next, we estimate the average firing rate ($R_i$) of each block in a unified SNN of T=6. Then, the energy consumption of a specified temporal configuration $t$ can be approximated as $\sum_{i=1}^{I} t_i \cdot R_i$. For example, the estimated energy consumption of a unified SNN with T=4 is calculated as $EC_4 = \sum_{i=1}^{I} 4R_i$. Based on this, we can obtain a group of temporal configurations with lower energy consumption ($EC$) for a given T=$T_g$, and we aim to identify the config with the maximum estimated accuracy from this set. This is formulated as the following optimization problem:

$$\text{maximize} \quad \text{ACC}_{\text{estimated}} = \sum_{i=1}^{I} K_i \sqrt{\log_2 t_i} + c$$

$$\text{s.t.} \quad \sum_{i=1}^{I} t_i \cdot R_i \leq EC_{T_g}$$

$$T_{min} \leq t_1, t_2, \ldots, t_l \leq T_{max},$$

where $t_i$ is the $i$-th component of temporal configurations $\boldsymbol{t}$ and $EC_{T_g}$ is the given uper bound of energy.

In order to solve the above problem, we adopt the depth-first search (DFS) algorithm to search in the solution space. To obtain a more accurate accuracy for each temporal configuration $t$, we perform three times of BN calibrations and take the average of the accuracies.

## A.10 Loss Landscapes of MTT and SDT

To visually confirm the flatter minimum achieved by the model trained with MTT, we trained two ResNet18 using SDT and MTT respectively on CIFAR100 for 300 epochs and plotted their loss landscapes in Fig. 10. The 1D landscape is plotted using the code provided by Li et al. (2018), whose basic idea is to view the trained weights as a high-dimension points and plots its surroundings. We observed that MTT led the model to a flatter minimum which indicates improved generalizability.

## A.11 Verifying Generalizability Through Gradient Metrics

Apart from noise injection, another famous metric that indicates the generalizability is the length of the gradient on weights $||\frac{\partial \mathcal{L}}{\partial W}||$ and the inputs $||\frac{\partial \mathcal{L}}{\partial x_i}||$. For $||\frac{\partial \mathcal{L}}{\partial W}||$, we evaluate the length of the gradient of loss over the entire training set for the convolution layers. For $||\frac{\partial \mathcal{L}}{\partial x_i}||$, we calculate the mean value of the length of each input gradient. The model trained by MTT exhibits a shorter gradient of both weights and inputs (see Tab. 11), which implies the model's strong robustness and generalizability.

**Table 11:** The gradient statistics of the model trained by SDT and MTT.

| Methods | $\|\|\frac{\partial \mathcal{L}}{\partial W}\|\|$ | $\|\|\frac{\partial \mathcal{L}}{\partial x_i}\|\|$ |
|---------|------|------|
| MTT | 11.59 | 1.81 |
| SDT | 38.08 | 7.78 |

**Table 12:** Results on seqMNIST.

| Methods | Acc |
|---------|-----|
| Our RNN | 56.22 |
| SNN SDT | 55.75 |
| SNN MTT | **64.56** |

**Table 13:** Results on Spiking Heidelberg Digits

| Methods | Acc |
|---------|-----|
| l=3 Repro by code of Hammouamri et al. (2023) | 75.26 |
| l=3 our SDT | 74.43 |
| l=3 our MTT | **79.68** |

**Table 14:** Results on Spiking Speech Commands

| Methods | Acc |
|---------|-----|
| Our SDT l=3 | 57.75 |
| Our MTT l=3 | **60.15** |

## A.12 Experiments on Audio and Sequential Datasets

Our research reveals that models trained by MTT can function effectively as a time encoder when the temporal configuration vectors used for training are monotonically non-increasing.

To illustrate this adaptability, we present the performance on three distinct temporal tasks: seqMNIST, Spiking Heidelberg Digits and Spiking Speech Commands.

For the sequential task seqMNIST, we utilized a simple fc LIF SNN with 2 hidden layers of width 64 and set the time constant $\tau$ to 0.99. We also trained an RNN with 2 hidden layers of width 64 for comparison. The results are as shown in Tab. 12.

For the Spiking Heidelberg Digits, we adopt the plain 3-layer feed-forward SNN architecture proposed by Cramer et al. (2020), a fully connected SNN with an input width of 70 and 128 LIF neurons in each of the 3 hidden layers. The timestep of the first layer is fixed to the input timestep, while the timesteps of subsequent layers are restricted to monotonically non-increasing. We set $\tau = 0.9753$, which is equivalent to the parameter $\lambda$ in the work of Cramer et al. (2020), namely $1 - 1/\tau$ in most other articles, and train the model for 150 epochs. To ensure the validity of our results, we also reproduce the result using the code provided by Hammouamri et al. (2023). The results are as shown in Tab. 13.

Spiking Speech Commands (SSC) (Cramer et al., 2020) is a spiking dataset converted from Google Speech Commands v0.2 and is tailored for SNN. For SSC, we continue using the same architecture and the same parameters as we used in SHD, except that here we only train the model for 60 epochs. The results are shown in Tab. 14.

## A.13 Computational Cost Analysis for MTT

**Time Complexity** When training with GPUs, the time required for a single forward and backward pass of normal SNNs is proportional to the time steps $T$. Therefore, the time cost for a normal SNN with $T$ time steps within a single iteration can be expressed as:

$$C(T) = T \cdot k$$

where $k$ is the time cost of a single time step. Now consider the cost of MTT. Before inserting the TTM module, the time cost at stage $i$ can be expressed as $C_i(T) = T \cdot k_i$, where $k = \sum_i k_i$. Let $C_{\text{TTM}}$ denote the total time cost of all TTM modules. The total time cost for one iteration of MTT can then be expressed as:

$$C_{\text{MTT}} = C_{\text{TTM}} + \sum_{i=1}^{s} \sum_{j=1}^{G} T_j^{(i)} \cdot k_j$$

Here, $s$ is the sampling times, and $G$ is the total number of stages. Note that BN calibration is only needed before inference, which requires only a few forward passes and incurs negligible overhead during training.

Due to the randomness in temporal configuration sampling, we calculate the expectation of the time cost under given $T_{\min}$ and $T_{\max}$ as follows:

$$E(C_{\text{MTT}}(T_{\min}, T_{\max})) = E(C_{\text{TTM}}(T_{\min}, T_{\max})) + s \cdot k \cdot \frac{T_{\min} + T_{\max}}{2}$$

Since a single TTM module involves at most $T_{\max}$ tensor multiplications and additions, its cost is negligible compared to the main model. After ignoring the $C_{\text{TTM}}$ term, the time cost expectation is:

$$E(C_{\text{MTT}}(T_{\min}, T_{\max})) \approx s \cdot k \cdot \frac{T_{\min} + T_{\max}}{2}$$

Thus, the time cost ratio between MTT and SDT is approximately:

$$\frac{s(T_{\min} + T_{\max})}{2T}$$

We verified this analysis by testing the first-epoch time of SDT and MTT on various datasets and models. All the experiments were conducted with RTX3090 GPUs and data-parallel training. The results are shown in Table 15.

**Table 15:** Experimental results of first-epoch training time for both MTT and SDT. MTT config is denoted by $s[T_{\min}, T_{\max}]$ where $s$ is sampling times each iteration, $[T_{\min}, T_{\max}]$ is the sampling range of $T$

| Model | Dataset | GPUs | Batch Size | MTT $s[T_{\min}, T_{\max}]$ | MTT Time | SDT $T$ | SDT Time | Actual | Theory |
|-------|---------|------|-----------|------|------|------|------|------|------|
| ResNet19 | CIFAR100 | 3 | 256 | 3[1, 6] | 193s | 6 | 109s | 1.77x | 1.75x |
| ResNet19 | CIFAR100 | 3 | 256 | 3[1, 10] | 328s | 10 | 190s | 1.73x | 1.65x |
| ResNet18 | CIFAR10-DVS | 2 | 50 | 3[1, 10] | 123s | 10 | 77s | 1.60x | 1.65x |

As shown, the experimental results align well with the theoretical analysis. According to our analysis, MTT's overhead is approximately 1.5 times that of SDT when $T$ is not too small.

**Space Complexity** MTT performs immediate backward passes after forward passes and accumulates gradients of all temporal configurations sampled within a single iteration. Because the computation graph and temporary tensors are instantly released after backpropagation, the theoretical maximal memory usage of MTT is comparable to standard SDT. However, since the maximal memory usage only occurs when the time steps of all stages are set to $T$, the intermediate memory usage of MTT may be smaller than SDT. We tested the GPU memory usage at the end of the first epoch, and the results are shown in Tab. 16. The experimental results confirm that MTT's memory usage is consistent with theoretical expectations.

**Table 16:** Experimental results of first-epoch memory usage for both MTT and SDT. $s[T_{\min}, T_{\max}]$ denotes MTT samples $s$ temporal configs each iteration, each time step is sampled from $[T_{\min}, T_{\max}]$

| Model | Dataset | GPUs | Method | MTT Memory (per GPU) |
|-------|---------|------|--------|------|
| ResNet18 | CIFAR10-DVS | 2 | MTT 3[1, 10] | 6640MiB |
| ResNet18 | CIFAR10-DVS | 2 | MTT 3[10, 10] | 7025MiB |
| ResNet18 | CIFAR10-DVS | 2 | SDT $T = 10$ | 7295MiB |

