# OpenReview forum: "Temporal Flexibility in Spiking Neural Networks: Towards Generalization Across Time Steps and Deployment Friendliness"
_ICLR.cc/2025/Conference — ICLR 2025 Poster_

### Official Review · Reviewer_ksDE · 2024-11-01

**Soundness:** 2
**Presentation:** 2
**Contribution:** 3
**Rating:** 6
**Confidence:** 4

**Summary:**

The manuscript introduces a new method, MTT, for training spiking neural networks (SNNs) that perform well across different numbers of timesteps during inference. These SNNs are referred to as Temporally Flexible SNNs (TFSNNs). The proposed method divides an SNN model into multiple stages, assigns each stage a random number of timesteps, and conducts multiple forward passes on the same data batch. The outputs are aggregated into a single loss function, and backpropagation through time (BPTT) is used to update the weights. Experimental results demonstrate that this approach enables the SNN model to perform consistently across varying timesteps during inference. Additionally, models trained using this method show enhanced generalization and robustness to noise, as well as high accuracy across multiple static and dynamic image classification datasets.

**Strengths:**

The manuscript identifies an interesting issue with direct training methods for SNNs: their limited performance when performing inference under a different timestep configuration than the one used during training. It proposes a solution that allows models to perform well across a range of timesteps, enhancing their flexibility and robustness. The models trained under this approach demonstrate improved generalization and competitive accuracy compared to SOTA direct training methods. Furthermore, the manuscript is overall well-written, with a clear and accessible description of the proposed method.

**Weaknesses:**

The problem definition and the proposed method's alignment with it remain somewhat unclear. The manuscript introduces the problem of "temporal inflexibility" in standard direct training methods, suggesting that this limitation could affect SNN deployment on fully event-driven hardware. However, the manuscript's focus is primarily on synchronous discrete models trained on static image datasets, with only a minor experiment (Table 5) dedicated to event-driven models, which lacks sufficient detail. Thus, the manuscript does not fully address the implications of "temporal inflexibility" for event-driven hardware deployment, instead centering on the need for inference performance consistency across multiple timesteps, a goal that may not directly relate to event-driven applications.

If the objective is to facilitate SNN deployment on event-driven hardware, the manuscript should clarify the differences between models described in Section 3.1 and fully event-driven models. For instance, Section 5.1 mentions that SPECK removes time-step-wise operations like clocked bias addition. How does the LIF model change after removing such operations? Conversely, if the goal is multi-timestep inference, further explanation is needed to justify its relevance for static image classification tasks. In these cases, where temporal information is absent, the optimal approach would be to achieve accurate inference with the minimum number of timesteps (ideally $T=1$, as in ANNs).

The experimental setup and results presentation require further clarification. For example, in Section 5.1, "Temporal Flexibility Across Time Steps", Table 2 is not referenced in the text, making it unclear what these results indicate. Additionally, in Table 3, which compares the proposed method with SOTA ANN-SNN methods, the settings for the ANN-SNN methods are unclear, was the conversion applied once for a single $T$ value, then tested across various $ T $ values, or was the conversion performed individually for each $ T $ value in Table 3? Similar issues with experimental descriptions and discussions are present in Sections 5.1 and 5.3.

**Questions:**

- What specific problem is the manuscript addressing, deployment on event-driven hardware or inference consistency across multiple timesteps? How does the proposed method align with this problem?
- What are the differences between the LIF models in Section 3.1 and those used in event-driven simulations?
- How does the method generalize to values of  $T $ outside the range used during training ($ [T_{\text{min}}, T_{\text{max}}] $) on event-based datasets?
- How was Figure 9 generated?
- What is the distinction between SDT and SDT* in Table 1?
- Could you clarify the "specific application scenarios" referenced in line 208?

---

> ### Author Response · Authors · 2024-11-23
> **Rebuttal of the Weakness 1 & Question 1**
>
> > The problem definition and the proposed method's alignment with it remain somewhat unclear. The manuscript introduces the problem of "temporal inflexibility" in standard direct training methods, suggesting that this limitation could affect SNN deployment on fully event-driven hardware. However, the manuscript's focus is primarily on synchronous discrete models trained on static image datasets, with only a minor experiment (Table 5) dedicated to event-driven models, which lacks sufficient detail. Thus, the manuscript does not fully address the implications of "temporal inflexibility" for event-driven hardware deployment, instead centering on the need for inference performance consistency across multiple timesteps, a goal that may not directly relate to event-driven applications.
>
> > What specific problem is the manuscript addressing, deployment on event-driven hardware or inference consistency across multiple timesteps? How does the proposed method align with this problem?
>
> **A:** Thank you for your critical and insightful feedback! We’ll respond to your concerns one by one:
>
> **1. Clarification of the Problem Addressed by Our Work**
>
> You mentioned that the specific problem our work addresses was unclear. Allow us to explicitly articulate it here:
>
> Our work aims to bridge the gap between SNN training and deployment on real hardware, mainly focusing on the gap between training and event-driven deployment. Current mainstream methods for efficiently training large-scale SNNs are based on time-stepped backpropagation. However, there exists a significant mismatch between the temporal structure of the network during training and deployment. This mismatch manifests in two ways:
> - For asynchronous event-driven hardware, the concept of discrete time steps does not exist. Time-stepped simulation during training is merely an approximation, and the temporal structure of real event-driven platforms differs substantially. Moreover, event-driven deployment is difficult to accelerate using GPUs.
> - For synchronous clock-driven hardware, the time step T is a tunable hyperparameter. During deployment, T may be adjusted to achieve trade-offs between energy efficiency and accuracy. For instance, recent studies have shown that dynamically adjusting T during inference can enhance performance-energy efficiency trade-offs.
>
> **2. About the Relationship between Time-stepped Experiments and the Problem**
>
> First, we sincerely apologize for any confusion caused by the extensive time-stepped experiments.
>
> We include these experiments because the training process still relies on time-stepped methods, even when training models for event-driven asynchronous deployment. Furthermore, fully event-driven asynchronous inference can be seen in a time-stepped perspective with a vast number of time steps T. We provide the proof in Sec. 3.2 and give an intuition here. When T is sufficiently high, at most one event occurs per time step, which effectively mirrors the event-driven paradigm, where the membrane potential of neurons is updated independently for the arrival of each event.
>
> *In the revised version of our manuscript, we have added new descriptions in Sec. 3.3 ("Event-Based Simulation, Event-Driven LIF/IF Model, and Hardware") to clarify the relationship between these two paradigms.*
>
> **3. Clarifications on the proposed method’s alignment with the problem**
>
> We have refined Sec. 3 and 4.1 in the revised manuscript to clarify the objectives of this work. Here, we provide a summary. We identify the temporal inflexibility caused by traditional training methods (referred to as standard direct training) as a key factor contributing to the training-inference gap. For event-driven deployment, the gap lies between GPU-accelerated time-stepped training and event-based inference similar to a high T time-stepped scenario. For clock-driven deployment, the gap is between training T and dynamic inference T. These methods optimize the model performance under a single temporal configuration, resulting in good performance during training but poor generalization to different temporal structures during inference. Our proposed approach addresses this by optimizing the model across multiple temporal configurations during training, effectively reducing temporal inflexibility and improving temporal flexibility. Remarkably, the experimental results demonstrate that SNNs trained with our method generalize better not only to low-T configurations but also to fully event-driven settings and high-T configs.
>
> **4. Details of Event-Driven Experiments**
>
> We apologize that the description of experiments in Table 5 was insufficiently detailed. Since the Table 5 experiments are among the most critical in our work, we have added comprehensive details in Sec. 3.3, 5.1, and Appendices A.2 and A.3. Additionally, the C++ code for the simulator used in these experiments is fully available in the supplementary materials. Please feel free to check it to ensure no details are missing.

---

> ### Author Response · Authors · 2024-11-23
> **Rebuttal of the Weakness 2 & Question 2**
>
> > If the objective is to facilitate SNN deployment on event-driven hardware, the manuscript should clarify the differences between models described in Section 3.1 and fully event-driven models. For instance, Section 5.1 mentions that SPECK removes time-step-wise operations like clocked bias addition. How does the LIF model change after removing such operations? Conversely, if the goal is multi-timestep inference, further explanation is needed to justify its relevance for static image classification tasks. In these cases, where temporal information is absent, the optimal approach would be to achieve accurate inference with the minimum number of timesteps (ideally T=1, as in ANNs).
>
> > What are the differences between the LIF models in Section 3.1 and those used in event-driven simulations?
>
> **A:** We apologize for not emphasizing the distinction between the neurons in Section 3.1 and fully event-driven models in the original manuscript. We have updated Sections 3.1, 3.2, and 3.3 in the revised version to address this issue. We provide a brief explanation here.
>
> In Section 3.1 of the original manuscript, we introduced the LIF neuron model, which was used for experiments on time-stepped models in this study. This was to facilitate fair comparisons with existing works. For event-based datasets, however, we employed the IF neuron model without decay to meet the requirements of fully event-driven neurons on SPECK hardware. From a neuronal perspective, the IF model can be considered a special case of the LIF model, where the decay constant (τ\tau) is set to 1. In this case, the membrane potential only updates upon the arrival of an event and does not decay over time. See Sec 3.2, 3.3 in the revised version for more details.
>
> Despite the dynamics differences between the LIF and IF models, our proposed method is effective for both. This is because our approach focuses on incorporating multiple temporal structures to enable flexibility in adapting to the temporal resolution of the input, independent of the specific internal dynamics of the neuron model.
>
> Additionally, since fully event-driven platforms do not support linear or convolutional layers with biases, the network architecture had to be adjusted accordingly. Due to page limitations in the main text and the fact that hardware-related constraint details are not the focus of this paper, we have included the relevant settings in Appendix A.3.

---

> ### Author Response · Authors · 2024-11-23
> **Rebuttal of the Weakness 3, Question 3, 4, 5, and 6**
>
> > The experimental setup and results presentation require further clarification. For example, in Section 5.1, "Temporal Flexibility Across Time Steps", Table 2 is not referenced in the text, making it unclear what these results indicate. Additionally, in Table 3, which compares the proposed method with SOTA ANN-SNN methods, the settings for the ANN-SNN methods are unclear, was the conversion applied once for a single T value, then tested across various T values, or was the conversion performed individually for each T value in Table 3? Similar issues with experimental descriptions and discussions are present in Sections 5.1 and 5.3.
>
> **A:** Thank you for your meticulous review and for pointing out that Table 2 was not referenced in the manuscript. We have addressed it in the revised version.
>
> Additionally, we have replaced the bottom part of Table 2 with a curve plot in Figure 4, which compares SDT and MTT. This new plot provides a more intuitive illustration of the overall higher performance trend achieved by MTT across various time steps.
>
> Regarding Table 3, all ANN-SNN conversion methods were performed individually for each T value and tested for the same T. In contrast, our method trains a single set of weights and evaluates across various T values. Compared to the SOTA conversion methods in Table 3—although they use fine-tuning tailored to each specific T—our approach achieves a universally robust model that maintains high performance across different T values. We have optimized the writing of the Table 3 section in the revised version to emphasize these results. For the experiments in Sections 5.1 and 5.3, we have provided detailed descriptions both in the corresponding subsections and in Appendices A.2 and A.3 of the revised manuscript.
>
> > How does the method generalize to values of T outside the range used during training on event-based datasets?
>
> **A:** After standard direct training (SDT), an SNN can still retain some accuracy when evaluated at T higher than the T used during training. However, its performance degrades significantly, showing poor generalization (as shown in Table 1 and Figure 4  of the revised version). On event-based datasets, larger T-values better approximate event-driven scenarios, but the maximum T that can be supported during training on GPUs is limited by memory constraints.
>
> The training method proposed in this work improves the temporal flexibility of the network by optimizing it across multiple temporal structures during training. This allows the network to generalize better to temporal structures different from those used during training, resulting in strong performance even at high T-values.
>
> > How was Figure 9 generated?
>
> **A:** Thank you for your suggestion. We indeed overlooked including the method for plotting Figure 9 (currently Figure 10 in the revised version). Figure 9 was generated using the method described in [1], which visualizes the loss landscape by adding perturbations to the weights. A flatter minimum in the loss landscape indicates a more generalized set of weights. We have added this detail in Appendix A.10 of the revised version.
>
> > What is the distinction between SDT and SDT* in Table 1?
>
> **A:** Apologies for the oversight. While we mentioned this in Section 4.1 of the manuscript, it was not clearly stated in the original figure caption. In the revised version, we have added the following clarification:
>
> *"SDT" denotes SNNs independently trained with SDT at each T. "SDT" denotes a single SNN trained at T=6 and inferred at other T.*
>
> > Could you clarify the "specific application scenarios" referenced in line 208?
>
> **A:** We apologize for the ambiguity. Here, "specific application scenarios" refer to situations where the sensor’s framing time steps vary depending on the application. For example, in autonomous driving on highways, low latency is critical, requiring the camera to achieve a high frame rate, i.e., a large number of framing time steps. On the other hand, for latency-insensitive tasks like gesture recognition, the camera’s frame rate requirement is much lower.
>
> The intended meaning of the statement is that “Temporal Flexibility” enables SNNs to perform universally well across different time steps, facilitating their application in various scenarios with different latency requirements. Typically, these scenarios demand high time steps, which are challenging to handle using the existing BPTT training paradigm.
>
> [1] Li H, Xu Z, Taylor G, et al. Visualizing the loss landscape of neural nets[J]. Advances in neural information processing systems, 2018, 31.

---

> > ### Comment · Reviewer_ksDE · 2024-11-25
> >
> > Thank you for your detailed response. Most of my concerns have been addressed, and I will be updating my score.
> >
> > However, I have one additional question for the authors. Could you elaborate on the energy efficiency of asynchronous event-driven hardware compared to synchronous event-driven hardware? Specifically, wouldn’t processing events individually in asynchronous hardware lead to increased energy consumption due to additional memory accesses? In contrast, synchronous hardware operates with finite time steps, potentially reducing the number of operations and memory accesses. A discussion on this aspect would provide valuable insights into the trade-offs between these two approaches and help better contextualize the contributions of the proposed method.

---

> ### Author Response · Authors · 2024-11-26
>
> We sincerely appreciate your valuable feedback and support for our work. Your review has provided significant insights that have greatly improved our manuscript. The discussion on the energy consumption differences between asynchronous and synchronous hardware is an important topic, and we are delighted to delve into this further with you.
>
>
> Firstly, it is worth noting that, as highlighted in [1], *"the main advantage of new SNN accelerators compared to ANNs on digital hardware comes primarily from exploiting the sparsity of spikes."* This sparsity is usually guaranteed when processing event streams generated by event sensors such as DVS cameras, which capture only pixels with significant luminance changes. Under such conditions, the dominant factor in energy consumption becomes the resting power of the chip [2].
>
>
> From our understanding, the key advantage of asynchronous chips over synchronous ones lies in their extremely low resting power. As noted in [2], *"The fully asynchronous architecture of Speck, which renders computing capacity solely dependent on input data, constitutes the key factor behind its persistent 'always-on' profile. In this paradigm, the neuromorphic chip no longer needs the global or local clock signal, efficiently preventing the redundant power consumed by clock empty flips."* In contrast, synchronous chips must maintain clock pulses and execute time-step-wise operations, such as bias accumulation, even without spike input.
>
>
> You raised an interesting point in your response: *"In contrast, synchronous hardware operates with finite time steps, potentially reducing the number of operations and memory accesses."* We agree that this scenario is generally relevant when processing static images. A typical approach involves repeating the static image $T$ times as input to the first convolutional (encoding) layer of the SNN, generating spike sequences for subsequent network computation. The finite $T$ reduces encoding loss, improving performance. However, this setup is more applicable when using frame-based cameras as the input source, which, unlike DVS cameras, cannot ensure sparsity.
>
>
> For example, in a warehouse monitoring application, a frame camera continuously produces images at a fixed frame rate for SNN processing, even when there are no changes in the scene. This can result in unnecessary operations. By contrast, a DVS camera only outputs events when pixel luminance changes occur, and it outputs only the affected pixels as event streams, ensuring sparsity.
>
>
> The energy efficiency advantages of asynchronous setups are further demonstrated in the comparative data provided in Table S1 of [2]'s appendix. While we cannot include images in this response due to OpenReview's limitations, we have excerpted some data for your reference below:
>
>
>
>
> | **Platform**     | **BrainScales** | **SpiNNaker** | **Neurogrid** | **TrueNorth** | **Darwin** | **Loihi** | **Loihi-2** | **Tianjic** | **Speck**     |
> |-------------------|-----------------|---------------|---------------|---------------|------------|-----------|-------------|-------------|---------------|
> | **Power**         | 1300mW          | 1000mW @180MHz| 150mW         | 63-300mW      | 58.8mW @1.8V+70MHz| 74mW    | N.A.         | 950mW@1.2V, 400mW@0.9V| 0.42-15mW @1.2V |
> | **Clock**         | Partially Async | Partially Async | Async         | Partially Async | Sync     | Partially Async | Partially Async | Sync       | Async         |
>
>
>
> [1] Dampfhoffer M, Mesquida T, Valentian A, et al. Are SNNs really more energy-efficient than ANNs? An in-depth hardware-aware study[J]. IEEE Transactions on Emerging Topics in Computational Intelligence, 2022, 7(3): 731-741.
>
>
> [2] Yao M, Richter O, Zhao G, et al. Spike-based dynamic computing with asynchronous sensing-computing neuromorphic chip[J]. Nature Communications, 2024, 15(1): 4464.

---

### Official Review · Reviewer_g9wC · 2024-11-02

**Soundness:** 3
**Presentation:** 3
**Contribution:** 3
**Rating:** 8
**Confidence:** 5

**Summary:**

This work first addresses the limitations of using fixed time steps in conventional SNN training, which lead to low generalization capability and performance degradation during practical deployment. Additionally, it highlights the challenges in dynamically balancing energy-performance trade-offs under this constraint. Starting from the Naive Mixture Training, this manuscript incrementally develops a framework that incorporates hybrid time-step and temporal flexibility training methods. By applying a strategic up/down rounding technique, the authors group the stages of networks like VGG and ResNet into multiple dynamic time-step segments for training. Notably, the paper presents results on neuromorphic hardware (Speck V2) and simulation results on asynchronous platforms, offering fresh insights into the practical advantages of dynamic time-step strategies.
The manuscript is logically clear and coherent, with the methodology presented in a gradual and systematic manner.
However, I still have a few questions and suggestions for modifications (and additions).

**Strengths:**

1. The manuscript is well-structured, logically coherent, and clearly articulated.

2. The figures are well-designed.

3. The defined spike difference has practical significance, offering a valuable metric for optimizing GPU training and on-chip deployment.

4. Experimental results demonstrate that the proposed MTT-enabled TFSNN outperforms the baseline in accuracy and exhibits stronger generalization capabilities.

**Weaknesses:**

1. In line 234 of the manuscript, the time step range is described as "$T_{max}$ to $G^{T_{max}}$" Should  $G^{T_{max}}$ be corrected to ${T_{max}}^{G}$ here?

2. What is the difference between $T=a_i$ and $T=t_i$ in Figure 1? Are they referring to different sample sets? If so, please clarify this in the caption.

3. The time step serves as a search space, which has a certain relationship with Neural Architecture Search (NAS). Please provide some discussion on the connection between the two.

4. Although it is understood that the focus of this paper is on reducing the gap between training and deployment, the sampling of samples and the grouped calculation of loss evidently increase training overhead. Please provide some discussion on this in the appendix.

**Questions:**

1. To my knowledge, the highest-performing architecture in the SNN field is the Spiking Transformer [1-5]. Please discuss whether the proposed method can be effectively applied to the Spiking Transformer. If feasible, please provide some preliminary experimental results.

[1] Zhou, Z., Zhu, Y., He, C., Wang, Y., Shuicheng, Y. A. N., Tian, Y., & Yuan, L. Spikformer: When Spiking Neural Network Meets Transformer. In The Eleventh International Conference on Learning Representations.

[2] Yao, M., Hu, J., Zhou, Z., Yuan, L., Tian, Y., Xu, B., & Li, G. (2024). Spike-driven transformer. Advances in neural information processing systems, 36.

[3] Zhou, C., Yu, L., Zhou, Z., Ma, Z., Zhang, H., Zhou, H., & Tian, Y. (2023). Spikingformer: Spike-driven residual learning for transformer-based spiking neural network. arXiv preprint arXiv:2304.11954.

[4] Zhou, Z., Che, K., Fang, W., Tian, K., Zhu, Y., Yan, S., ... & Yuan, L. (2024). Spikformer v2: Join the high accuracy club on imagenet with an snn ticket. arXiv preprint arXiv:2401.02020.

[5] Yao, M., Hu, J., Hu, T., Xu, Y., Zhou, Z., Tian, Y., ... & Li, G. Spike-driven Transformer V2: Meta Spiking Neural Network Architecture Inspiring the Design of Next-generation Neuromorphic Chips. In The Twelfth International Conference on Learning Representations.

**Details Of Ethics Concerns:**

None.

---

> ### Author Response · Authors · 2024-11-27
> **Rebuttal of the Weakness 1, 2, 3, and 4**
>
> >In line 234 of the manuscript, the time step range is described as “$T_{max}$ to $G^{T_{max}}$”, should $G^{T_{max}}$ be corrected to $T_{max}^{G}$
>
> **A:** Thank you for your careful review! Here, $G^{T_{max}}$ indeed should be corrected to $T_{max}^{G}$. We apologize for our typo and have corrected this issue in the latest version.
>
> >What is the difference between $T=a_i$ and $T=t_i$ in Figure 1? Are they referring to different sample sets? If so, please clarify this in the caption.
>
> **A:** Thank you for your suggestion! We agree that $a$ and $t$ here may cause some confusion and have edited the caption and Figure 1 to clarify. In the latest version of our paper, we use $t_i^{(j)}$ to denote the time step of the $i$-th stage in the $j$-th sampled time config.
>
> >The time step serves as a search space, which has a certain relationship with Neural Architecture Search (NAS). Please provide some discussion on the connection between the two.
>
> **A:** Thanks for the question! This is an interesting topic, and we are delighted to discuss it with you. From the perspective of NAS, MTT is very similar to the one-shot NAS search for different branches (each path has a unique time step). Nevertheless, it is indeed the case that MTT and NAS are not entirely consistent. One of the most significant differences between MTT and one-shot NAS lies in the shared architectures and weights of different branches in MTT. Plus, the objectives of MTT and NAS are different. While NAS aims to find the optimal substructure, the goal of MTT is to acquire an SNN that adapts to all different time steps.
>
> Out of curiosity, however, we also conducted additional research to analyze the performance of MTT-trained networks under different time-step configurations. Details of this study can be found in Appendix A.8. Specifically, we developed a method to predict the accuracy of an MTT-trained model for a given time-step configuration. This method can also evaluate the contribution of each stage's time steps to the overall accuracy of the configuration. Our study found that the impact of time steps on accuracy varies significantly across different stages, which indicates that the importance of time steps differs among the stages of an SNN.
>
> > Although it is understood that the focus of this paper is on reducing the gap between training and deployment, the sampling of samples and the grouped calculation of loss evidently increase training overhead. Please provide some discussion on this in the appendix.
>
> **A:** Thank you for your thoughtful feedback! We have included a new section in the revised version which thoroughly analyzes MTT's training costs theoretically and then validates the theory experimentally. See our **response to reviewer wNg9’s comment 1 or appendix section A.13 Computational Cost Analysis for MTT.**

---

> ### Author Response · Authors · 2024-11-27
> **Rebuttal of the Question 1**
>
> Thank you for your question! As you mentioned, spike-based transformers have gained significant attention in the SNN field in recent years. This architecture enables SNNs to achieve performance comparable to traditional ANNs on static datasets. Notably, models like Spikformer V2 [4] and Spike-driven Transformer V2 [5] have reached 80% accuracy on the ImageNet dataset with only 4 inference time steps. This synergy between SNNs and transformers has opened new possibilities for large-scale SNNs on static datasets. For a more comprehensive review, we included discussions on spike-based transformer architectures [1-5] in the related works section.
>
> Currently, most mainstream SNN transformers adopt Spiking Self-Attention (SSA) or similar structures, where spiking attention usually involves the multiplication of spikes within the same timestep. However, generalizing SSA-like structures to event-driven scenarios, especially those with high timesteps, presents challenges. This is probably because, with the same spiking input, increasing the number of timesteps (T) causes spikes within a single timestep to become sparse, making it difficult for spikes to interact with each other and preventing SSA from generating spikes effectively.
>
> To evaluate the effectiveness of our method on such structures, we built a SpikFormer-2-256, which consists of two encoder blocks and one SPS module. Following the typical settings for event-driven datasets in this paper, we first trained the model on CIFAR10-DVS using SDT for 100 epochs and then fine-tuned it for 30 epochs using both SDT and MTT. Both $T$ and $T_{\text{max}}$ were set to 10. $T_{min}$ was lifted to 5. We observed that removing bias during fine-tuning made it difficult for SDT to converge. Therefore, we retain biases in this experiment to ensure reasonable comparison. The results are in the table below. Our experiments demonstrate that MTT can enhance the temporal flexibility of the model, even for structures that struggle to generalize across different time steps.
>
> | T   | 1     | 5     | 10    | 25    | 50    | 75    | 100   |
> |-----|-------|-------|-------|-------|-------|-------|-------|
> | MTT | **12.4**  | **49.09** | 61.39 | **25.4**  | **17.04** | **13.81** | **11.29** |
> | SDT | 9.98  | 12.02 | **61.49** | 24.29 | 15.02 | 10.99 | 10.58 |
>
> [1] Zhou, Z., Zhu, Y., He, C., Wang, Y., Shuicheng, Y. A. N., Tian, Y., & Yuan, L. Spikformer: When Spiking Neural Network Meets Transformer. In The Eleventh International Conference on Learning Representations.
>
> [2] Yao, M., Hu, J., Zhou, Z., Yuan, L., Tian, Y., Xu, B., & Li, G. (2024). Spike-driven transformer. Advances in neural information processing systems, 36.
>
> [3] Zhou, C., Yu, L., Zhou, Z., Ma, Z., Zhang, H., Zhou, H., & Tian, Y. (2023). Spikingformer: Spike-driven residual learning for transformer-based spiking neural network. arXiv preprint arXiv:2304.11954.
>
> [4] Zhou, Z., Che, K., Fang, W., Tian, K., Zhu, Y., Yan, S., ... & Yuan, L. (2024). Spikformer v2: Join the high accuracy club on imagenet with an snn ticket. arXiv preprint arXiv:2401.02020.
>
> [5] Yao, M., Hu, J., Hu, T., Xu, Y., Zhou, Z., Tian, Y., ... & Li, G. Spike-driven Transformer V2: Meta Spiking Neural Network Architecture Inspiring the Design of Next-generation Neuromorphic Chips. In The Twelfth International Conference on Learning Representations.

---

> ### Comment · Reviewer_g9wC · 2024-11-27
>
> I greatly appreciate the author's efforts in addressing my concerns, and I am willing to accept an increase in my score.

---

> > ### Author Response · Authors · 2024-11-28
> >
> > We sincerely appreciate your decision to increase your score and are truly grateful for your careful review, valuable suggestions, and positive feedback, all of which have greatly enhanced the quality of our work.

---

### Official Review · Reviewer_spLP · 2024-11-02

**Soundness:** 3
**Presentation:** 2
**Contribution:** 3
**Rating:** 6
**Confidence:** 4

**Summary:**

This paper identifies the temporal flexibility problem for spiking neural networks (SNNs) that is important for SNNs’ deployment on time-step-free fully event-driven chips, and proposes a novel mixed time-step training method to alleviate the problem under current direct training approaches. For evaluation, the trained models are tested under both time-step-based and fully event-driven settings, where the latter includes both the Speck chip and a developed simulator. Experiments show promising performance for temporal flexibility, robustness, deployment to the fully event-driven setting, and commonly used static and neuromorphic datasets.

**Strengths:**

1. This paper considers an important problem in bridging algorithms of SNNs and the real deployment on fully event-driven neuromorphic hardware. It is significant for real applications.

2. Experiments are comprehensive, covering not only the commonly adopted GPU simulation settings but also real chips or a similar event-driven simulator. Various datasets including static images, neuromorphic images, and audio, have been considered to verify the good performance of the proposed method.

3. Results show that the proposed method can achieve near SOTA performance and has much better performance when deployed under the fully event-driven setting.

**Weaknesses:**

1. The paper lacks sufficient details for the considered fully event-driven setting. For example, what are the details of the Speck chip and the developed simulator? How is input or output formulated, and how does asynchronization influence the network? This can affect some claims, for example, “large-scale SNNs on fully event-driven scenarios”, since only N-MNIST is verified on the real chip and other experiments are on the simulator. These details should be included to enable justification if simulator experiments can support the claim.

2. For the presentation, there are some not fully discussed logical gaps.

First, there is a gap between the identified temporal inflexibility problem and deployment on fully event-driven chips, because the former is still in the time-step-based setting while the latter is in the time-step-free setting. It is better to add more explanations about why the considered flexibility under the synchronized setting can certainly improve time-step-free settings, e.g., why flexibility can alleviate the problem caused by asynchronization.

Second, the motivation from NMT to MTT is missing.

Third, there is no formal and rigorous definition for temporal flexibility. Even for SNNs trained with a specific T, they can naturally run for different time steps, just with a drop in performance. To what extent can a model be called flexible or inflexible? For the proposed method, there is also a performance drop and the improvement is to reduce it rather than introducing a new property. The concept is mainly a quantitative comparison instead of a qualitative one, so I think it is not rigorous to claim the proposed method to “exhibit temporal flexibility”.

3. This paper mainly focuses on empirical verifications while theoretical analysis is limited.

**Questions:**

To alleviate the influence of time steps, will it be better to consider the combination with some online-through-time training methods with instantaneous losses [1,2] rather than backpropagation through time, since the former can naturally consider information from different time steps and may be suitable for on-chip training to better adapt to real settings?

[1] Online training through time for spiking neural networks. NeurIPS, 2022.

[2] A solution to the learning dilemma for recurrent networks of spiking neurons. Nature Communications, 2020.

---

> ### Author Response · Authors · 2024-11-25
> **Rebuttal of the Weakness 1**
>
> >The paper lacks sufficient details for the considered fully event-driven setting. For example, what are the details of the Speck chip and the developed simulator? How is input or output formulated, and how does asynchronization influence the network? This can affect some claims, for example, “large-scale SNNs on fully event-driven scenarios”, since only N-MNIST is verified on the real chip and other experiments are on the simulator. These details should be included to enable justification if simulator experiments can support the claim.
>
> **A:** Thank you very much for your professional suggestions. We sincerely apologize for the lack of relevant details in our original submission, partly due to page limitations and our simulator, which we plan to present in full detail in our following work. We will now address each of your concerns in detail:
>
> Speck [1] is a fully asynchronous event-driven commercial chip that integrates both a DVS camera and computing units. In the computational part of Speck, the input and output are formulated as event streams, which are encoded in Address-Event Representation (AER). The "asynchronization" itself does not directly alter the architecture of the neural network, but when weights trained on GPUs using a synchronous paradigm are directly deployed to an asynchronous chip, we observe a significant drop in network performance. In this work, we identify this specific issue and propose a potential solution, MTT, to mitigate this problem.
> Due to the limited capacity of Speck, which only supports a 9-layer convolutional network structure and relatively narrow network widths (limited number of channels), we were only able to deploy smaller models on it to evaluate their performance on the N-MNIST dataset.
>
> To test larger models on more challenging datasets, we developed a custom C++ asynchronous event-driven simulator. This simulator faithfully implements the asynchronous operators found in Speck, such as the asynchronous event-driven convolution mentioned in [2], with minimal deviation from the real chip's output (as detailed in the main text Section 5.1). Therefore, it effectively emulates the behavior of the asynchronous chip. Additionally, the simulator supports a time-step-based setting, ensuring that with the same time-step configuration (that's to say, the same T), the outputs of a model are perfectly aligned between the simulator and the GPU, even down to a character-by-character match, which strongly validates the correctness of the simulator's code implementation.
>
> With this simulator, we were able to evaluate larger models on more challenging datasets, such as DVS-CIFAR10. As shown in Table 5 of the original paper, we increased the scale of the network by upgrading the backbone to VGGSNN (which significantly increased the parameter count). Our proposed method demonstrated the ability to alleviate the performance drop observed during deployment.
>
> [1] Richter O, Xing Y, De Marchi M, et al. Speck: A smart event-based vision sensor with a low latency 327k neuron convolutional neuronal network processing pipeline[J]. arXiv preprint arXiv:2304.06793, 2023.
>
> [2] Yao M, Richter O, Zhao G, et al. Spike-based dynamic computing with asynchronous sensing-computing neuromorphic chip[J]. Nature Communications, 2024, 15(1): 4464.

---

> ### Author Response · Authors · 2024-11-25
> **Rebuttal of the Weakness 2**
>
> >For the presentation, there are some not fully discussed logical gaps.
>
> >First, there is a gap between the identified temporal inflexibility problem and deployment on fully event-driven chips, because the former is still in the time-step-based setting while the latter is in the time-step-free setting. It is better to add more explanations about why the considered flexibility under the synchronized setting can certainly improve time-step-free settings, e.g., why flexibility can alleviate the problem caused by asynchronization.
>
> **A:** Thank you for your valuable suggestions! We have added a more in-depth explanation of the relationship between the fully event-driven model and the time-step-based model in Sections 3.1, 3.2, and 3.3 of the revised version, and we provide a summary here.
>
> First, there is a mathematical connection between the event-driven framework and the time-step-based framework. In fact, the model used for event-driven deployment still needs to be trained using the time-step-based framework to leverage GPU acceleration. The rationale behind this is that, when the input consists of instantaneous spikes, the time-step-based inference serves as a low-precision approximation of the event-driven inference. The event-driven inference can be seen in a time-stepped perspective with a vast number of time steps (T), as derived mathematically in Sec. 3.2 and Sec. 3.3. When the time steps are sufficiently fine-grained, at most one event occurs within each time step, which effectively equates to the event-driven paradigm, where each event independently updates the neuron's membrane potential. Networks with strong temporal flexibility are not only better suited to scenarios where T is smaller than that used during training, but can also generalize to cases with extremely high T, which underlies their suitability for asynchronous deployment.
>
> > Second, the motivation from NMT to MTT is missing.
>
> **A:** Thank you for your kind reminder. Upon review, we realized that we indeed missed the motivation for transitioning from NMT to MTT. We have now added the following content in Sec. 4.3 and refined the explanation of NMT in Sec. 4.1 and Sec. 4.2.
>
> “... The success of NMT lies in its incorporation of diverse temporal structures during training. A straightforward idea to improve is to include more temporal structures. However, the number of temporal structures in NMT scales linearly with Tmax, and excessively large T cannot be trained on current GPUs. To introduce more temporal structures while keeping Tmax not increasing, we propose Mixed Timestep Training (MTT). …”
>
> > Third, there is no formal and rigorous definition for temporal flexibility. Even for SNNs trained with a specific T, they can naturally run for different time steps, just with a drop in performance. To what extent can a model be called flexible or inflexible? For the proposed method, there is also a performance drop and the improvement is to reduce it rather than introducing a new property. The concept is mainly a quantitative comparison instead of a qualitative one, so I think it is not rigorous to claim the proposed method to “exhibit temporal flexibility”.
>
> **A:** Thank you for your suggestions. We acknowledge that our descriptions of temporal flexibility and temporally flexible SNN were imprecise, and we apologize for any confusion caused. Here, we clarify these concepts:
>
> First, we provide a more precise definition of temporal flexibility: it refers to a model's ability to generalize to temporal structures other than those used during training, measured by its performance on unseen configurations compared to models trained specifically on those configurations.
>
> Second, we revise our statement on temporally flexible SNNs. As you correctly pointed out, SDT-trained networks retain some accuracy on other temporal structures. However, overfitting to a single time step significantly degrades their performance on different temporal structures. MTT-trained models effectively mitigate this issue. Despite the huge temporal flexibility improvement, the obtained models are still not ideal temporally flexible SNNs that achieve comparable performance across all temporal structures as models specifically trained for each structure. Nonetheless, MTT is still a key step toward achieving fully temporal flexible SNNs.
>
> We have updated the manuscript to clarify the descriptions related to temporal flexibility and temporally flexible SNNs.

---

> ### Author Response · Authors · 2024-11-25
> **Rebuttal of the Weakness 3 and Question 1**
>
> >This paper mainly focuses on empirical verifications while theoretical analysis is limited.
>
> **A:** In Sections 3.2 and 3.3 of the revised version, we have added detailed explanations and theoretical analysis of the time-stepped simulation and event-based simulation, establishing the connection between the two. Since the main contribution of this paper lies in identifying the issues faced during SNN deployment, providing a new perspective on the relationship between event-driven models and time-stepped models, and, for the first time, evaluating the performance of SNNs in an asynchronous environment, MTT is proposed as a possible empirical solution to address the "temporal inflexibility." Due to page limitations and the focus of this paper, we have not conducted an extensive theoretical analysis of MTT but have provided some discussion in Section 4.2 under the Network Generalization part.
>
> > To alleviate the influence of time steps, will it be better to consider the combination with some online-through-time training methods with instantaneous losses [1,2] rather than backpropagation through time, since the former can naturally consider information from different time steps and may be suitable for on-chip training to better adapt to real settings?
>
> **A:** Thank you for sharing the two works! We had previously investigated similar studies. However, while OTTT and methods with instantaneous loss consider information across different time steps, their training is still conducted on a single $T$ or $\Delta t$ / $dt$. Such approaches thus fail to address overfitting to a single time step within a temporal simulation framework, as illustrated in Figure 4. It is evident that TET, whose loss is the same as instantaneous loss, also suffers from overfitting to a single time step. Furthermore, OTTT neglects temporal gradients, potentially limiting its ability to extract temporal information, whereas our method retains these gradients.
>
> That said, we believe combining the idea of multi-temporal-structure training with online training is a highly promising and exciting direction. We have discussed this in Section 6 (Conclusion) of the revised version with works [1, 2] included to provide a more comprehensive perspective on the on-chip deployment of SNNs.

---

> > ### Comment · Reviewer_spLP · 2024-11-26
> >
> > I would like to thank the authors for the detailed response. From the explanation, the event-driven setting is based on calculation with precise spiking timestamps, while the time-step setting requires discretization in the time, and this difference leads to the performance gap, right? And the ''temporal flexibility'' may be viewed as a kind of property describing the robustness to different discretization settings, so more robustness may lead to better generalization under the precise timestamp calculation, is that right? If so, I would suggest describing it more precisely in the context of discretization, which differs from using more time steps under the same discretization setting. That is, different discretization settings means using different discretization intervals so that there are different discrete time steps for an equivalent total time, while using more time steps with the same discretization interval leads to a longer total time. The ''temporal structure'' may be better understood as ''discretization structure'', and ''temporal flexibility'' as ''discretization flexibility''.
> >
> > If the flexibility refers to discretization $\Delta t$, then the parameters in the calculation of SNNs should depend on $\Delta t$, such as $\tau$, $W$, etc. Is it considered in experiments?

---

> ### Author Response · Authors · 2024-11-27
>
> >If so, I would suggest describing it more precisely in the context of discretization, which differs from using more time steps under the same discretization setting.
>
> **A:** Thank you for your valuable suggestions. After reviewing your comment, I believe we have no fundamental disagreement on the main direction, but perhaps some misalignment in understanding specific details. Please allow us to explain the reason why we use the term “temporal flexibility”.
>
> First, “temporal flexibility”, compared to “discretization flexibility”, is more accurate for static datasets and thus is a more general concept. For DVS datasets and event-driven platforms, temporal flexibility does correspond to "using different discretization intervals so that there are different discrete time steps for an equivalent total time" as your description. However, in the case of static datasets and synchronized hardware, it’s hard to define the concept of “discretization of time” since the static data do not have timestamps or any temporal attributes. Therefore, the “temporal flexibility” we propose here is a broader notion, referring to the strong adaptability of SNNs in both time-stepped frameworks and event-driven platforms, as demonstrated in Table 3 and Figure 4 of the paper.
>
> Second, “temporal flexibility” can better express the issue identified in this paper- “temporal inflexibility”, which means the model overfits to a specific T and suffers from performance degradation when changing to another temporal structure. This issue encompasses both static and DVS datasets simultaneously. While this paper mainly focuses on deployment on fully event-driven platforms, the proposed method also alleviates the temporal inflexibility of time-stepped inference. For instance, in dynamic time-step settings, our approach ensures universally high performance across varying numbers of timesteps.
>
> Besides, the name “temporal flexibility” better reflects the nature of the training problem for deployment on *existing* fully event-driven hardware. Existing fully asynchronous event-driven chips care more about the relative order of events because they mainly support IF neurons [1], whose membrane potential changes only upon event arrival and does not decay over time. In this situation, the spiking timestamps only determine the sequence of event arrivals but do not impact the computation itself. When training networks in a time-stepped setting, the event stream is divided into T bins, and events within the same frame are treated as simultaneous, which disrupts the original temporal order of events. From this perspective, temporal flexibility better captures the essence of this issue.
>
>
> >If the flexibility refers to discretization $\Delta t$, then the parameters in the calculation of SNNs should depend on $\Delta t$, such as $\tau$, $W$, etc. Is it considered in experiments?
>
> **A:** Your question is highly relevant and critical. In fact, when designing the training and deployment experiments for our event-driven model, we carefully considered how various parameters change with simulation time steps. We are delighted to discuss this topic further.
>
> According to our derivation, the decay parameter $\tau$ is indeed related to $\Delta t$ and may need to be adjusted with changes in simulation time steps. In Section 3.2, we established the relationship between the LIF neuron differential equation model and time-stepped simulation, where we derived $\tau = 1 - \Delta t / \tau_0$, with $\tau_0$ representing the time constant of the LIF neuron differential equation.
>
> In our asynchronous experiments, the value of $\tau$ strictly adheres to this relationship. As mentioned in Section 3.3, modern fully event-driven asynchronous chips primarily support event-driven IF neurons. Our simulator also uses event-driven IF neurons to align with these neuromorphic chips. Based on the derivations in Sections 3.1 and 3.2, both event-driven and time-stepped IF neurons can be viewed as limiting cases of LIF neurons as $\tau_0 \to +\infty$. In this case, $\lim_{\tau_0 \to +\infty} \tau = \lim_{\tau_0 \to +\infty} (1 - \Delta t / \tau_0) = 1$. Therefore, we set $\tau = 1$ in the time-stepped training framework, namely using time-stepped IF neurons during training.
>
> We hope this explanation addresses your question. As for other parameters, such as $W$, our derivation did not reveal any direct interaction with $\Delta t$.
>
>
> [1] Yao M, Richter O, Zhao G, et al. Spike-based dynamic computing with asynchronous sensing-computing neuromorphic chip[J]. Nature Communications, 2024, 15(1): 4464.

---

### Official Review · Reviewer_cDVC · 2024-11-03

**Soundness:** 3
**Presentation:** 2
**Contribution:** 3
**Rating:** 6
**Confidence:** 3

**Summary:**

The paper proposes the Mixed Time - step Training (MTT) method, which is utilized to train Spiking Neural Networks, thereby achieving generalization across distinct time steps. This method empowers SNNs to accommodate diverse temporal structures. Furthermore, the paper conducts validation experiments to confirm the effectiveness of MTT. The method devises specific loss, a temporal transformation module, and network partitioning to fulfill the above-mentioned objective.

**Strengths:**

The paper presents a promising SNN training approach, namely MTT, which attains remarkable outcomes in terms of temporal adaptability and model performance, possessing definite innovation and practical value. The co-training method involving multiple time steps is both rational and effective. The experiments are comprehensive, having been conducted on diverse datasets, and there are also tests on event - driven neuromorphic systems.

**Weaknesses:**

1.The writing in this paper has room for improvement. The authors ought to place greater emphasis on the key points throughout the article. For instance, concerning the temporal flexibility across time steps, the experiment should stress that different time steps perform well in general. Avoid delving too deeply into which particular time step works well, as this may obscure the aim of the work. The authors should focus on presenting the overall performance trend across various time steps instead of detailing the results for each separate time step. This would enhance the emphasis on the key point of generalization across time steps.

2. Some additional details regarding the experiment and the method are required. Specifically, an explanation must be given as to why the time steps in Table 4 are represented as decimals. Moreover, the meanings of the symbols used should be clearly elucidated.

**Questions:**

Could you provide a clear definition of V(t) when it is first introduced in section 3.1, and explain its significance in the context of the neuron model？

---

> ### Author Response · Authors · 2024-11-22
> **Rebuttal of the Weakness 1, Weakness 2 and Question 1.**
>
> We appreciate your recognition of our paper and your valuable suggestions. Regarding your suggestions, we have revised the corresponding sections of the paper, and the updated parts are colored in blue. The modifications are summarized below.
>
> ---
> > The writing in this paper has room for improvement. The authors ought to place greater emphasis on the key points throughout the article. For instance, concerning the temporal flexibility across time steps, the experiment should stress that different time steps perform well in general…
>
> **A**: Thank you for your thoughtful feedback and suggestions on our manuscript. We appreciate your insights and agree that emphasizing the key points more clearly will enhance the overall quality of the paper. In response to your suggestion, we have revised the corresponding sections.
>
> Firstly, we have replaced the bottom part of Table 2 with a curve plot showing the comparison between SDT and MTT in Figure 4, which gives a more intuitive demonstration of the overall higher performance trend across various time steps.
>
>
> Secondly, for a conspicuous emphasis, we have added a new statement at the beginning of Section 5.3:
>
> “*While our work mainly focuses on improving the temporal flexibility of networks, the models trained by MTT maintain a performance on par with other SOTA methods.*”
>
> and have deleted the original sentence in the end:
>
> “*The models trained by MTT not only maintain a performance close to SOTA methods but also exhibit considerable temporal flexibility.*”
>
> ---
>
> > Some additional details regarding the experiment and the method are required. Specifically, an explanation must be given as to why the time steps in Table 4 are represented as decimals. Moreover, the meanings of the symbols used should be clearly elucidated.
>
> **A**: We apologize for the missing detail of the experiment related to Table 4. We have supplemented the relevant details in this part. And the reason why the time steps shown in Table 4 are represented as decimals is because they represent the average required time steps for all samples of the test set. During inference, we employ the SEENN [1] method, which stops the SNN's inference upon reaching a specific condition rather than ceasing after completing all time steps.
>
> ---
>
> > Could you provide a clear definition of V(t) when it is first introduced in section 3.1, and explain its significance in the context of the neuron model？
>
> **A**: We apologize for the confusion caused by our oversight. In the original manuscript, V(t) represents the temporary membrane potential used for calculating spike generation and reset. We have added the definition of V(t) in the revised version.
>
> [1] Li Y, Geller T, Kim Y, et al. Seenn: Towards temporal spiking early exit neural networks[J]. Advances in Neural Information Processing Systems, 2024, 36.

---

> > ### Comment · Reviewer_cDVC · 2024-11-28
> >
> > Thank you for your detailed response. Most of my concerns have been addressed. And I greatly appreciate the author's efforts in improving the paper. This version is better to understand.

---

> > > ### Author Response · Authors · 2024-11-28
> > >
> > > We are truly grateful for your careful review, valuable suggestions, and positive feedback, all of which have greatly improved the quality of our work.

---

### Official Review · Reviewer_wNg9 · 2024-11-04

**Soundness:** 3
**Presentation:** 3
**Contribution:** 2
**Rating:** 5
**Confidence:** 4

**Summary:**

This paper introduces a training method, Mixed Time-step Training (MTT), aimed at improving the temporal flexibility of Spiking Neural Networks (SNNs) for more versatile deployment. The training method allows SNNs to adapt across diverse time steps by assigning random time steps to different stages of the network in each iteration. After training, TFSNNs are deployed and evaluated on both time-step-based and event-driven platforms. The authors compare their method on static and DVS datasets with ANN-SNN conversion methods (Jiang et al.2023) and SEENN (Li et al., 2023), emphasize its generalization to different time steps.

**Strengths:**

1.	The paper introduces a novel training method to address the side effects of temporal
inflexibility caused by the prevailing training paradigms.
2.	The paper conducts intensive experiments on GPU, neuromorphic chips, and event-
driven simulator to testify to its effectiveness.

**Weaknesses:**

1. The proposed method aims to enhance the model's temporal flexibility through mixture training. However, the authors do not provide a clear analysis of the training complexity or computational costs involved. Table 9 highlights the relationship between the sampling frequency and training epochs, yet further details are needed to elucidate these aspects comprehensively.
2. Some performance improvements reported by the authors appear less substantial upon closer examination. For example, in Table 4, the addition of MTT has a very limited effect on improving overall performance. Similarly, in the generalization comparison involving Gaussian noise-injected inputs (Figure 5), while the accuracy of the MTT method consistently exceeds that of SDT, the margin is minimal compared to the overall drop in accuracy as noise intensity increases. These observations make it challenging to substantiate the claim that the model’s generalization is significantly enhanced.

**Questions:**

Please refer to the weaknesses section. Additionally, there are two more questions:
1. The authors suggest that their models can infer across various time steps without additional fine-tuning. However, whether flexibility across all time steps is necessary, especially outside of event-driven scenarios, remains an open question. Given the potential complexity of the proposed training approach (see Weakness 1 for details), it may be more efficient and focused to fine-tune the model for specific possible platforms rather than attempting universal temporal flexibility.
2.The authors compare their method to the current state-of-the-art (SOTA) in ANN-to-SNN conversion (Table 3) and report improved performance when T is low. However, performance declines slightly as T increases. Likewise, in Table 4, although naïve mixture training demonstrates some advantage over standard direct training at smaller T values, this benefit diminishes as T approaches 5 or 6. This raises an question: given that SNNs have limited capacity to capture temporal dynamics across time steps when T is very small, is this improvement practically significant?

---

> ### Author Response · Authors · 2024-11-17
> **Response to Reviewer wNg9 (Part 1/5)**
>
> Comment 1:
>
>
> The proposed method aims to enhance the model's temporal flexibility through mixture training. However, the authors do not provide a clear analysis of the training complexity or computational costs involved. Table 9 highlights the relationship between the sampling frequency and training epochs, yet further details are needed to elucidate these aspects comprehensively.
>
>
>
> ---
>
>
>
> Response to comment 1:
>
>
> Thank you for your valuable suggestions. Analyzing the computational overhead during training is indeed crucial. In response, we will add a section (**appendix section A.13**) to provide comprehensive theoretical analysis and experimental validation of MTT's training costs.
>
> **Computational Cost Analysis for MTT**
>
> **Time Complexity** When training with GPUs, the time required for a single forward and backward pass of normal SNNs is proportional to the time steps $T$. Therefore, the time cost for a normal SNN with $T$ time steps within a single iteration can be expressed as
> $$C(T) = T \cdot k$$
>
> where $k$ is the time cost of a single time step. Now consider the cost of MTT. Before inserting the TTM module, the time cost at stage $i$ can be expressed as $C_i(T) = T \cdot k_i$, where $k = \sum_i k_i$. Let $C_{TTM}$ denote the total time cost of all TTM modules. The total time cost for one iteration of MTT can then be expressed as:
>
> $$
> C_{MTT} = C_{TTM} + \sum_{i=1}^s \sum_{j=1}^{G} T_{j}^{(i)} \cdot k_j
> $$
>
> Here, $s$ is the sampling times, and $G$ is the total number of stages. Note that BN calibration is only needed before inference, which requires only a few forward passes and incurs negligible overhead during training.
>
> Due to the randomness in temporal configuration sampling, we calculate the expectation of the time cost under given $T_{\text{min}}$ and $T_{\text{max}}$ as follows:
>
> $$
> E(C_{MTT}(T_{\text{min}}, T_{\text{max}})) = E(C_{TTM}(T_{\text{min}}, T_{\text{max}})) + s \cdot k \cdot \frac{T_{\text{min}} + T_{\text{max}}}{2}
> $$
>
> Since a single TTM module involves at most $T_{\text{max}}$ tensor multiplications and additions, its cost is negligible compared to the main model. After ignoring the $C_{TTM}$ term, the time cost expectation is
>
> $$
> E(C_{MTT}(T_{\text{min}}, T_{\text{max}})) \approx s \cdot k \cdot \frac{T_{\text{min}} + T_{\text{max}}}{2}
> $$
>
> Thus, the time cost ratio between MTT and SDT is approximately:
>
> $$
> \frac{s(T_{\text{min}} + T_{\text{max}})}{2T}
> $$
>
> We verified this analysis by testing the first-epoch time of SDT and MTT on various datasets and models. All the experiments are conducted with RTX3090 GPUs and data-paralleled. The results are shown in the table below:
>
> **Table C1.1 Experimental results of first-epoch training time for both MTT and SDT. MTT config is denoted by $s[T_{\text{min}}, T_{\text{max}}]$ where $s$ is sampling times each iteration, $[T_{\text{min}}, T_{\text{max}}]$ is the sampling range of $T$**
>
> | Model    | Dataset     | GPUs | Batch Size | MTT $s$[$T_{\text{min}}$, $T_{\text{max}}$] | MTT Time | SDT $T$ | SDT Time | Actual | Our Theory |
> | - | - | - | - | - | - | - | - | - | - |
> | ResNet19 | CIFAR100    | 3    | 256        | 3[1, 6]                                     | 193s     | 6       | 109s     | 1.77x  | 1.75x      |
> | ResNet19 | CIFAR100    | 3    | 256        | 3[1, 10]                                    | 328s     | 10      | 190s     | 1.73x  | 1.65x      |
> | ResNet18 | CIFAR10-DVS | 2    | 50         | 3[1, 10]                                    | 123s     | 10      | 77s      | 1.60x  | 1.65x      |
>
> As shown, the experimental results align well with the theoretical analysis. According to our analysis, MTT’s overhead is approximately 1.5 times that of SDT when $T$ is not too small.
>
> **Space Complexity** MTT performs immediate backward passes after forward passes and accumulates gradients of all temporal configurations sampled within a single iteration. Because the computation graph and temporary tensors are instantly released after backpropagation, the theoretical maximal memory usage of MTT is comparable to standard SDT. However, since the maximal memory usage only happens when the time steps of all stages are set to T, the intermediate memory usage of MTT may be smaller than SDT. We tested the GPU memory usage at the end of the first epoch, and the results are as follows:
>
> **Table C1.2 Experimental results of first-epoch memory usage for both MTT and SDT. $s[T_{\text{min}}, T_{\text{max}}]$ denotes MTT samples $s$ temporal configs each iteration, each time step is sampled from $[T_{\text{min}}, T_{\text{max}}]$**
>
> | Model    | Dataset     | GPUs | Method        | MTT Memory (per GPU) |
> | - | - | - | - | - |
> | ResNet18 | CIFAR10-DVS | 2    | MTT 3[1, 10]  | 6640MiB              |
> | ResNet18 | CIFAR10-DVS | 2    | MTT 3[10, 10] | 7025MiB              |
> | ResNet18 | CIFAR10-DVS | 2    | SDT $T=10$    | 7295MiB              |
>
> The experimental results confirm that MTT's memory usage is consistent with theoretical expectations.

---

> > ### Comment · Reviewer_wNg9 · 2024-11-17
> >
> > Regarding time complexity, the response provides an analysis and experimental evaluation of the computational cost per epoch. However, how does this impact the convergence behavior? How does the overall training time compare to SDT under these circumstances?

---

> ### Author Response · Authors · 2024-11-17
> **Response to Reviewer wNg9 (Part 2/5)**
>
> Comment 2:
>
> Some performance improvements reported by the authors appear less substantial upon closer examination. For example, in Table 4, the addition of MTT has a very limited effect on improving overall performance. Similarly, in the generalization comparison involving Gaussian noise-injected inputs (Figure 5), while the accuracy of the MTT method consistently exceeds that of SDT, the margin is minimal compared to the overall drop in accuracy as noise intensity increases. These observations make it challenging to substantiate the claim that the model’s generalization is significantly enhanced.
>
>
>
> ---
>
>
>
> Response to Comment 2:
>
> We sincerely apologize for the confusion caused by our paper, and we greatly appreciate you pointing out these concerns. We will address each of your points one by one.
>
> First, we must emphasize that improving GPU accuracy at a specific time step was never the primary goal of this work. The major aim of our approach is to enhance the SNNs across time steps, thereby reducing the gap between SNN training and practical deployment. The improvement in network generalization, distinct from generalization across timesteps also known as temporal flexibility, is merely an additional benefit of MTT.
>
> In **Table 4**, we compare our model with the original SEENN model. The reason the improvements "appear" to be less significant is primarily due to two factors:
>
> 1. The accuracy of the model at this stage is already above 96%, so further improvements become inherently more difficult.
> 2. SEENN uses a training method that is not solely based on SDT but rather TET, as we explain in line 357 of the original manuscript. This further demonstrates that our method is better suited for dynamic time-step inference compared to the TET method.
>
> When input noise is injected, the performance difference between MTT and SDT is indeed relatively small. However, to minimize the impact of random fluctuations, we conducted 5 independent experiments for each level of noise (as mentioned in lines 407, 409 of the manuscript), and in **Figure 4, 5**, the shaded areas represent the accuracy range (maximum and minimum) across the 5 trials. As shown in Figure 5, the widths of both red and blue shaded areas are very small, indicating that our results are relatively stable and not random.
>
> To further substantiate the claim of improved generalization, we have included another experiment in the appendix, where we measure network generalization using the **gradient norm** (see **Appendix Table 11**). It is evident that networks trained with MTT exhibit smaller gradient norms for both input and weights compared to networks with the same architecture trained using SDT, indicating that MTT converges to a flatter minima. We hope this clarifies your concerns.

---

> ### Author Response · Authors · 2024-11-17
> **Response to Reviewer wNg9 (Part 3/5)**
>
> Question 1:
>
> The authors suggest that their models can infer across various time steps without additional fine-tuning. However, whether flexibility across all time steps is necessary, especially outside of event-driven scenarios, remains an open question. Given the potential complexity of the proposed training approach (see Weakness 1 for details), it may be more efficient and focused to fine-tune the model for specific possible platforms rather than attempting universal temporal flexibility.
>
>
>
> ---
>
>
>
> Response to Question 1:
>
> Thank you for your question. We hope the following responses will help address your concerns.
>
> - **Although the question tends to focus on non-event-driven scenarios, we must be clear that event-driven scenarios are significant to SNNs.** In fact, as pointed out in [1], SNNs can only achieve significantly lower energy consumption than ANNs if they benefit from the sparsity of events, which indicates that event-driven scenarios are essential to fully leverage the energy efficiency advantages of SNNs.
> - **Temporal flexibility is important for event-driven scenarios.** The precise forward process in event-driven platforms is difficult to parallelize or accelerate using GPUs because of the sequential nature of events. Although time-stepped simulations fit in GPU-based frameworks like PyTorch, they are prone to overfitting to a specific time step. The proposal of temporal flexibility successfully bridges the gap and allows for fast training of event-driven friendly SNNs.
>
> - **Even for time-stepped/clock-driven scenarios, MTT is efficient.** In our response to Weakness 1, we analyzed the complexity of MTT. When the number of time steps ($T_{max}$) is not too small, the time overhead is approximately 1.5 times that of SDT, whose $T=T_{max}$. By spending only half more computational cost, we obtain a more universal model. This is clearly more efficient than fine-tuning a dedicated model for each specific time step.
> - **Direct fine-tuning for some scenarios, such as the high-T scenarios, is not practical. However, models trained by MTT can generalize to these scenarios.** For example, training at high T (e.g., T = 1000, 10000) is impractical within a time-stepped GPU framework because both the memory and computation costs of time-stepped training increase proportionally as T grows. As a result, both GPU memory usage and training time become thousands of times larger and unaffordable. The models trained by MTT, according to our experiments, can perform well even with very large T, as detailed in **Table 3** and **Table Q2.2**.
> - **SNNs with temporal flexibility enable on-chip dynamic adjustment of T (see Table 4), while fine-tuning for each T on chip is impossible.**
> - **The significance of temporal flexibility also lies in its connection with neuron dynamics simulation.** The neuron dynamics is usually discretized into time steps in recent studies, allowing SNNs to be trained using the BPTT paradigm on GPU devices effectively. However, the dynamics of spiking neurons are originally described by continuous differential equations. From this perspective, T can be seen as a dynamics-independent hyperparameter, and the network weights trained should be irrelevant to the choice of this hyperparameter. Therefore, “temporal flexibility” is a step towards the intrinsic dynamics of the original SNN.

---

> ### Author Response · Authors · 2024-11-17
> **Response to Reviewer wNg9 (Part 4/5)**
>
> Question 2:
>
> The authors compare their method to the current state-of-the-art (SOTA) in ANN-to-SNN conversion (Table 3) and report improved performance when T is low. However, performance declines slightly as T increases. Likewise, in Table 4, although naïve mixture training demonstrates some advantage over standard direct training at smaller T values, this benefit diminishes as T approaches 5 or 6. This raises an question: given that SNNs have limited capacity to capture temporal dynamics across time steps when T is very small, is this improvement practically significant?
>
> Response to Question 2:
>
> Thank you for your question! We will explain each point in order.
>
> - The comment mentioned that the performance of TFSNN in **Table 3** slightly decreases as T increases.
>   - However, after carefully reviewing **Table 3**, we did not observe this trend. The performance of TFSNN actually improves as T increases, whereas the ANN-SNN conversion methods show a degradation in performance with larger values of T. This performance drop may be due to the fact that these latest network conversion methods perform post-conversion fine-tuning for each T to improve accuracy at low time steps. In contrast, our TFSNN does not involve any fine-tuning when changing T. This further supports the claim that MTT-trained TFSNN exhibits a considerable degree of temporal flexibility.
>   - For convenient reference, we have pasted the original table from the manuscript below. Notes that we added the same data augmentation as the other two methods for fair comparison (see around line 353).
>
> **Table 3 Compare with SOTA ANN-SNN conversion methods on CIFAR100, ResNet18. $T_{max} = 6$ is used for MTT**
>
> | Method       | T=1       | T=2       | T=4       | T=8       | T=16      | T=32      | T=64      |
> | ------------ | --------- | --------- | --------- | --------- | --------- | --------- | --------- |
> | QCFS [2]     | -         | 70.29     | 75.67     | 78.48     | **79.48** | **79.62** | **79.54** |
> | SlipReLU [3] | 71.51     | 73.91     | 74.89     | 75.40     | 75.41     | 75.30     | 74.98     |
> | MTT          | **72.09** | **76.54** | **78.47** | **78.90** | 79.17     | 79.25     | 79.42     |
>
> - The comment also claimed that in Table 4, the performance of NMT shows significant improvement primarily when T is small, and then expressed concern that if improvements are only evident for small T, the practical significance of these results might be limited.
> - This is possibly referring to **Table 1**, as **Table 4** presents comparisons with SEENN using MTT and does not include any content related to NMT.
>   - First, it is important to clarify that the main objective of our work has never been to simply improve the accuracy of SNNs at a specific time step. Rather, the focus is to enable SNNs to break free from the constraints of training time steps. The experiment in **Table 1** clearly demonstrates the feasibility of this goal—*even the simplest NMT allows the SNN to generalize to other time steps*.
>   - Furthermore, the NMT method in **Table 1** was initially proposed as the simplest approach while we explored how to train TFSNNs. Our final method, MTT, was gradually developed and refined from the NMT framework. We have provided extensive ablation experiments on SDT → NMT → MTT, which thoroughly demonstrate the effectiveness of our approach (see **Figure 6**).
>   - A series of experiments show that networks trained with MTT not only maintain model performance as T decreases (see **Table 2**, **Table 4**, **Table 6**, **Table 8**), **but also preserve performance even when T is very high** (see **Table 3**). This, from another perspective, explains why TFSNNs are so well-suited for asynchronous deployment (see **Table 5**). When T is extremely high, most time frames have either no events or only a single event. In such cases, time-stepped inference becomes very similar yet not identical to the scenario of asynchronous chips, where events are sequentially passed into neurons.
>   - To further strengthen this argument, we have added another new experiment. We train VGGSNNs on CIFAR10-DVS by MTT and SDT, respectively, and then test their performance at extremely high time steps. Since the PyTorch-based framework cannot afford inference at this many time steps, we test them on our self-developed simulator. As shown in the below table, the performances get closer when T grows and eventually approximate the event-driven case.
>
> **Table Q2.2 Test accuracy of VGGSNNs trained by MTT/SDT with extremely large time step**
>
> |      | T=10 | T=1000 | T=100000 | Fully Event-driven |
> | ---- | ---- | ------ | -------- | ------------------ |
> | MTT  | 75.2 | 61.7   | 60.1     | 58.5               |
> | SDT  | 74.7 | 52.3   | 50.6     | 48.4               |

---

> ### Author Response · Authors · 2024-11-17
> **Response to Reviewer wNg9 (Part 5/5)**
>
> **References**
>
> [1]Dampfhoffer M, Mesquida T, Valentian A, et al. Are SNNs really more energy-efficient than ANNs? An in-depth hardware-aware study[J]. IEEE Transactions on Emerging Topics in Computational Intelligence, 2022, 7(3): 731-741.
>
> [2] Bu, Tong, et al. "Optimal ANN-SNN conversion for high-accuracy and ultra-low-latency spiking neural networks." *arXiv preprint arXiv:2303.04347* (2023).
>
> [3] Jiang, Haiyan, et al. "A unified optimization framework of ANN-SNN conversion: towards optimal mapping from activation values to firing rates." *International Conference on Machine Learning*. PMLR, 2023.

---

> ### Author Response · Authors · 2024-11-18
>
> Thank you for your question!
>
> The convergence behavior of training methods is crucial to the training time, and we appreciate the opportunity to discuss this further. Our experiments show that the convergence behavior of MTT is similar to that of SDT, and in many cases, the loss of MTT decreases even slightly faster. To demonstrate this, we have posted a table comparing the loss values of SDT and MTT across the same epochs during training, in the common setting of ResNet18 on CIFAR100 with 300 epochs as used in this paper. To make the comparison fair, we averaged the losses across the $s$ temporal configurations in a single iteration of MTT and compared this average to SDT. From the table, MTT achieves slightly faster loss convergence.
>
> Considering the provided analysis of the per-epoch computational cost and the experimental evaluation, it is straightforward to conclude that the overall training time of MTT is comparable to SDT.
>
> | Method | Epoch 0 | Epoch 50 | Epoch 100 | Epoch 150 | Epoch 200 | Epoch 250 |
> | ------ | ------- | -------- | --------- | --------- | --------- | --------- |
> | SDT    | 4.1141  | 0.7656   | 0.5470    | 0.3424    | 0.1685    | 0.03255   |
> | MTT    | 4.1239  | 0.7085   | 0.4715    | 0.2835    | 0.1170    | 0.0265    |

---

> > ### Comment · Reviewer_wNg9 · 2024-11-18
> >
> > I have a general question about the motivation behind your methods, which I find somewhat confusing.
> >
> > You claim that your method, particularly its application to neuromorphic chips, is beneficial. However, current models of SNNs based on these chips are typically used for traditional deep learning tasks, such as image classification, where the goal is often to minimize the dynamics characteristics. These models tend to use very few time steps, often only four or five, to simulate dynamics, to the extent that many such SNNs no longer exhibit spiking behaviors.
> >
> > Given that mainstream SNN tasks already utilize *very few time steps*, almost to the point where the dynamic characteristics are obscured, what is the necessity of introducing varied time steps in your training method?
> > Moreover, the scenarios that might genuinely benefit from your approach are more biologically plausible networks, where preserving neurodynamics is crucial, thus requiring many time steps for accurate simulation. However, your current work does not seem to address these types of networks at all.
> >
> > I feel that your method might be misapplied, focusing on contexts where its advantages are less impactful and overlooking potential applications where it could be truly advantageous.

---

> > > ### Author Response · Authors · 2024-11-19
> > >
> > > Thank you for your response. We perceive there might be a minor misunderstanding about our work. We would like to clarify that our work primarily identifies the practical challenges associated with deploying SNNs on asynchronous chips. There is a huge performance degradation when SNNs (trained on GPUs under a synchronous paradigm) are deployed on asynchronous chips. This decline is attributed to the disparities in running rules between synchronous clock-driven SNNs (which have the concept of the time step) and asynchronous event-driven SNNs (which lack the concept of the time step). As for the scenario of using neuromorphic chips you mentioned, we would like to clarify that these scenarios and chips you referred to are synchronous clock-driven. The operational rules of SNNs on these chips are almost identical to those on GPUs. So considering that traditional static images do not have temporal information, and even a single time step can input the complete information into the network, SNNs are capable of using extremely low time steps.
> > >
> > > Another type of neuromorphic chip is the asynchronous chip, which primarily receives event streams from DVS cameras instead of images. Asynchronous chip is event-driven without any hardware clock inside. When neurons are only active when they receive spikes. Hence, an asynchronous chip shows the extremely low power consumption advantage. For example, the power consumption of Speck of the Synsense ([1]) is at the milliwatt level, significantly lower than that of GPUs or synchronous chips.
> > >
> > > Besides the low power consumption, asynchronous chips also exhibit lower latency in asynchronous scenarios. Consider a task where a DVS camera captures an event stream from an action, and the SNN needs to infer the type of the action. Synchronous chips require a preprocessor (such as a CPU) to preprocess the DVS stream and reconstruct it into multiple image frames, and then send image frames into the deployed SNN for clock-driven inference. In contrast, asynchronous chips can simultaneously collect events from DVS cameras and directly process the event stream on the deployed SNN for event-driven inference, hence achieving lower latency [2]. Therefore, under these asynchronous scenarios, asynchronous chips are more efficient than synchronous chips.
> > >
> > > In fact, asynchronous chips and asynchronous scenarios are very important for the SNN field. However, current mainstream research focuses on synchronous scenarios, neglecting the problem when deploying SNNs on asynchronous chips. Our work identifies the problem termed “temporal inflexibility” in deployment and provides a method to acquire a high-performance asynchronous SNN from GPUs to alleviate that. (As we mentioned in a previous response: “When T is extremely high, most time frames have either no events or only a single event. In such cases, time-stepped inference becomes very similar to the scenario of asynchronous chips, where events are sequentially passed into neurons.”)
> > >
> > > We believe that this work has made a contribution to the development and application of the SNN field. And we hope that our work will inspire more research into asynchronous chips and scenarios and ultimately lead to more efficient and powerful asynchronous SNNs.
> > >
> > > Thanks again for your response. We hope that our response has addressed your concerns.
> > >
> > > [1] Richter O, Xing Y, De Marchi M, et al. Speck: A smart event-based vision sensor with a low latency 327k neuron convolutional neuronal network processing pipeline[J]. arXiv preprint arXiv:2304.06793, 2023.
> > >
> > > [2] Yao M, Richter O, Zhao G, et al. Spike-based dynamic computing with asynchronous sensing-computing neuromorphic chip[J]. Nature Communications, 2024, 15(1): 4464.

---

> ### Author Response · Authors · 2024-11-28
>
> As the discussion deadline approaches, we kindly ask if you could review our response and reconsider our work, if it is convenient for you. We are happy to address any additional questions and would be pleased to provide further clarification. It would be a pleasure to resolve any concerns you may have. Thank you for your time and valuable feedback!

---

> ### Comment · Reviewer_wNg9 · 2024-12-02
>
> Thank you for addressing my technical concerns in detail. However, I am not fully convinced due to what I perceive as *overclaims* in the title, contributions, and main theme of the paper.
>
> (1) The current title, "Temporal Flexibility in Spiking Neural Networks: Towards Generalization Across Time Steps and Deployment Friendliness," seems to target all types of Spiking Neural Networks (SNNs), including both ANN-SNN , which typically use minimal time steps with limited neurodynamics, and biologically plausible models, which employ more extensive time-stepping and richer neurodynamics. However, this work only involves a small subset of SNNs, yet claims applicability to the broader SNN spectrum. This generalization appears overstated and should be more accurately reflected in the manuscript.
>
> (2) As you have mentioned, this work primarily identifies the practical challenges associated with deploying SNNs on asynchronous chips. From my understanding, the concept of "Temporal Flexibility" is primarily applicable when transferring models from GPUs (or synchronous chips) to asynchronous chips. It is well known that asynchronous chips, due to their immature ecosystem and developmental complexities, have limited utility compared to GPUs. Given that most research groups still utilize GPU platforms, could it be that your method has limited applicability in current settings? While it is commendable to develop new methods for asynchronous chips—a promising direction that fosters innovation—it is crucial NOT TO OVERSTATE the scope of your method, especially at this stage.
>
> For these reasons, I have raised my score to a 5, acknowledging the resolution of my technical concerns. However, I strongly recommend that you moderate the claims and more precisely delineate the contribution to better reflect its significance to the field.

---

### Author Response · Authors · 2024-11-17
**General Response 1**

We sincerely appreciate all the reviewers for their valuable comments and suggestions on our work. In this general response, we aim to clarify again the motivation behind this study and its contributions to the field of spiking neural networks (SNNs).

Due to their significantly lower energy consumption compared to artificial neural networks (ANNs) on neuromorphic hardware, SNNs have garnered extensive attention. This paper focuses on addressing challenges related to SNN deployment on hardware rather than improving GPU performance for a specific timestep setting. This is because high GPU performance does not guarantee similar efficiency on neuromorphic devices.

In recent years, many studies have adopted the approach of discretizing the dynamic equations of spiking neurons using a hyperparameterized simulation timestep \(T\) to enable the backpropagation through time (BPTT) paradigm for training large-scale SNNs on GPUs. Rethinking this process, the ultimate goal of training SNNs on GPUs is to find effective weights for **deployment on neuromorphic devices with their complex dynamics**, rather than simply optimizing weights hyperparameterized by \(T\) for the discretized SNN.

In this context, our paper introduces a novel concept, "temporal flexibility," which describes the ability of SNNs to perform well across various simulation timesteps \(T\). We demonstrate the significant benefits of temporal flexibility for SNNs in dynamic timestep-based scenarios and asynchronous settings. To train a temporally flexible SNN, we build upon Native Mixture Training (NMT) and propose the Mixed Timestep Training (MTT) method. Through extensive experiments, we validate the effectiveness of MTT and highlight its advantages for SNN deployment.

MTT partially addresses the question of how to train an SNN with temporal flexibility, offering a foundation for future exploration. Additionally, it sheds light on minimizing the "performance gap between synchronous training and asynchronous deployment," thereby promoting research into practical SNN deployment.

---

### Meta-Review · Area_Chair_JgKi · 2024-12-19

**Metareview:**

Towards the temporal inflexibility issue of existing SNN, this paper explores the feasibility of training SNNs that generalize across different time steps and introduces a mixed time-step training strategy. Experiments demonstrate the effectiveness. After the rebuttal, it receives one borderline reject, three borderline accept, and one accept. The response well addresses most of the reviewers' concerns. The strength of the paper, including the clear motivation, interesting ideas, extensive experiments, and good results, are well recognized. I agree with them and think the current manuscript meets the requirements of this top conference. Reviewer wNg9 proposes an issue about the overclaim. I think it can be addressed in the revision.  Please incorporate the suggestion to moderate the claims and more precisely delineate the contribution to better reflect its significance to the field.

**Additional Comments On Reviewer Discussion:**

The response well addresses most of the reviewers' concerns. Reviewer wNg9 is still concerned about the overclaim issue, which I think can be addressed in the revision. I think the current manuscript meets the requirements of this top conference.

---

### Decision · Program_Chairs · 2025-01-22

Accept (Poster)